# MEDITAB: SCALING MEDICAL TABULAR DATA PREDICTORS VIA DATA CONSOLIDATION, ENRICHMENT, AND REFINEMENT

## ABSTRACT

Tabular data prediction has been employed in medical applications such as patient health risk prediction. However, existing methods usually revolve around the algorithm design while overlooking the significance of data engineering. Medical tabular datasets frequently exhibit significant heterogeneity across different sources, with limited sample sizes per source. As such, previous predictors are often trained on manually curated small datasets that struggle to generalize across different tabular datasets during inference. This paper proposes to scale medical tabular data predictors (`MediTab`) to various tabular inputs with varying features. The method uses a data engine that leverages large language models (LLMs) to consolidate tabular samples to overcome the barrier across tables with distinct schema. It also aligns out-domain data with the target task using a "learn, annotate, and refinement" pipeline. The expanded training data then enables the pre-trained `MediTab` to infer for arbitrary tabular input in the domain without fine-tuning, resulting in significant improvements over supervised baselines: it reaches an average ranking of 1.57 and 1.00 on 7 patient outcome prediction datasets and 3 trial outcome prediction datasets, respectively. In addition, `MediTab` exhibits impressive zero-shot performances: it outperforms supervised XGBoost models by $8.9\%$ and $17.2\%$ on average in two prediction tasks, respectively.

## 1 INTRODUCTION

Tabular data are structured as tables or spreadsheets in a relational database. Each row in the table represents a data sample, while columns represent various feature variables of different types, including categorical, numerical, binary, and textual features. Most previous papers focused on the *model* design of tabular predictors, mainly by (1) augmenting feature interactions via neural networks (Arik & Pfister, 2021), (2) improving tabular data representation learning by self-supervised pre-training (Yin et al., 2020; Yoon et al., 2020; Bahri et al., 2022), and (3) performing cross-tabular pre-training for transfer learning (Wang & Sun, 2022b; Zhu et al., 2023). Tabular data predictor was also employed in medicine, such as patient health risk prediction (Wang & Sun, 2022b), clinical trial outcome prediction (Fu et al., 2022), modeling Electronic Health Record (EHR) data for multitask learning (Hur et al., 2023), and unifying heterogeneous EHRs via text embeddings (Hur et al., 2022). Additionally, LLMs have been shown to be able to sample synthetic and yet highly realistic tabular data as well (Borisov et al., 2022; Theodorou et al., 2023).

Despite these significant advances, it is worth noting that the *data-centric* approaches have received comparatively less attention in prior research. Some prominent examples lie in the detection and mitigation of label noise (Wang et al., 2020; Northcutt et al., 2021), but they only address a fraction of the challenges in medical tabular data prediction. As illustrated in Figure 1, there is typically substantial heterogeneity among different data sources in medical data, and within each data source, the available sample sizes are small. Harnessing multi-source data requires extensive manual effort in terms of data cleaning and formatting. As such, current medical tabular prediction methods are often built on small handcrafted datasets and, hence, do not generalize across tabular datasets.

In this paper, we embrace a data-centric perspective to enhance the scalability of predictive models tailored for medical tabular data. Our core aim revolves around training a single tabular data predic-

tor to accommodate inputs with diverse feature sets. Technically, our framework, namely `MediTab`, encompasses three key components: *data consolidation*, *enrichment*, and *refinement* modules:

- **Data consolidation and enrichment** involves consolidating tabular samples with varying features and schemas using natural language descriptions. We also expand the training data by distilling knowledge from large language models and incorporating external tabular datasets.
- **Data refinement** rectifies errors and hallucinations introduced during the consolidation and enrichment stages. It also aligns a diverse set of tabular samples with the target task through a distantly supervised pipeline.

As illustrated in Figure 1, `MediTab` offers the advantages:

- **Multi-task learning and prediction**: the model can learn from and make predictions for multiple medical tabular datasets without requiring modifications or retraining.
- **Few-shot and zero-shot learning**: the model can quickly adapt to new prediction tasks using only a small amount of training data or even make predictions for any new tabular input when no training data is available.

In Section 2, we provide a detailed description of our approach. We present the experimental results in Section 3, where we demonstrate the effectiveness of our method on 7 patient outcome prediction datasets and 3 trial outcome prediction datasets, achieving an average performance ranking of **1.57** and **1.00**, respec-

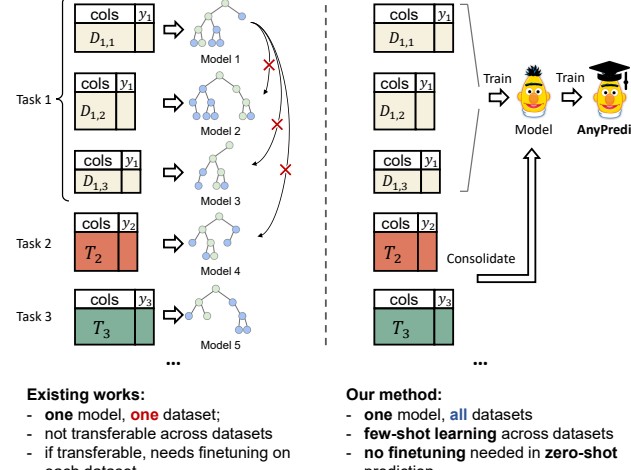

Figure 1: `MediTab` vs. existing tabular prediction methods. Existing methods learn and predict on a per-dataset basis, while `MediTab` can use data from the target task and all other tasks to improve performance.

tively, across tabular prediction baselines. Furthermore, our method shows impressive few-shot and zero-shot performances that are competitive with supervised baselines: the **zero-shot** `MediTab` outperforms supervised XGBoost by **8.9%** and **17.2%** on average in two prediction tasks, respectively. We discuss related work in Section 4 and conclude our findings in Section 5.

## 2 METHOD

### 2.1 PROBLEM FORMULATION

We characterize tabular prediction tasks by *dataset* $\mathbf{D}$ and *task* $\mathbf{T}$, where a task $\mathbf{T} = \{\mathbf{D}_1, \mathbf{D}_2, \dots\}$ consists of multiple *in-domain* datasets with varying features and schema but the same target label. For example, the patient mortality prediction task contains samples from many clinical trials (where input features differ between trials). For $\mathbf{T}_1$, the datasets from other tasks $\mathbf{T}_2, \mathbf{T}_3, \dots$ are considered *out-domain* since they differ in prediction objectives. As illustrated by Figure 1, existing methods for tabular prediction fall short in transfer learning across datasets, as each model learns from a single dataset $\mathbf{D}$ and needs to learn from scratch when encountering new datasets. On the contrary, `MediTab` extends the training data to all available tasks $\mathcal{T} = \{\mathbf{T}_1, \mathbf{T}_2, \dots\}$, demonstrating its flexibility to encode and predict for arbitrary tabular samples. After training, it serves all $\mathbf{D} \in \mathbf{T}_1$ without further fine-tuning. Our method eliminates the need to keep as many models as datasets, paving the way for the efficient and streamlined deployment of tabular prediction models. Depending on the use case, the problems that our method can handle can be classified into the following categories.

**Problem 1** (**Multi-task learning (MTL)**). *MTL is a machine learning technique where a single model is trained to perform multiple tasks simultaneously. Define $f : \mathcal{X} \mapsto \mathcal{Y}$ as a model that takes*

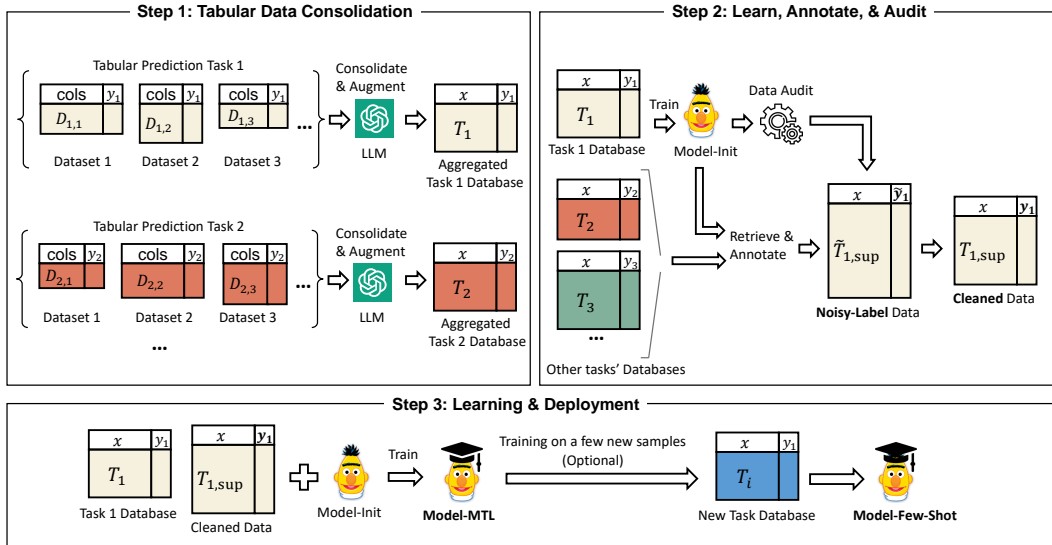

Figure 2: The demonstration of scaling medical tabular data predictors models (`MediTab`). It encompasses three steps: **Step 1** consolidates tabular datasets using LLM; **Step 2** aligns out-domain datasets with the target task; **Step 3** facilitates the predictor with cleaned supplementary data. More details are presented in Section 2.2.

*a consolidated tabular sample $x$ as input and predicts the target label $y$. The training dataset is formed by combining all the tabular inputs in $\mathbf{D}_* \in \mathbf{T}$. Once trained, the model $f$ is fixed and can be used to make predictions on any new samples $x \sim \mathbf{D}$, $\forall \mathbf{D} \in \mathbf{T}$.*

**Problem 2 (Zero-shot/Few-shot learning).** *The model $f$ is trained on $\mathbf{T} = \{\mathbf{D}_1, \dots, \mathbf{D}_N\}$. Then, it makes predictions for a new dataset $\mathbf{D}_{N+1}$ that has not been included in the training data. Model $f$ performs zero-shot learning if no label is available for all samples in $\mathbf{D}_{N+1}$; Model $f$ performs few-shot learning to predict for $\mathbf{D}_{N+1}$ if a few labeled samples are available.*

## 2.2 THE MEDITAB FRAMEWORK

As illustrated in Figure 2, our method consists of:

**Step 1: Tabular Data Consolidation**. The tabular datasets $\mathbf{D}$ differ in their features, schema, and particularly in their target objectives if they are from distinct tasks $\mathbf{T}$. The consolidation is accomplished by converting each row of the table into natural language descriptions that consider the data schema. This conversion process transforms all tabular data into text data that share the same semantic space, enabling them to be utilized in language modeling. Additionally, we can produce diverse consolidated samples by describing one sample in multiple different ways, which allows for data augmentation. To prevent hallucinations that may occur during this transformation, an audit module that utilizes LLMs is employed to perform self-check and self-correction. Our goal of patient survival classification is the same for each dataset; however, we use a diverse number of datasets, so the task is indeed different.

**Step 2: Learn, Annotate, & Audit**. Our method can benefit from out of domain datasets $\mathbf{T}_* \in \mathcal{T}$ through our annotation and data importance pipeline. Once it is trained on $\mathbf{T}_1$, it is used to produce pseudo labels for samples from all other tasks, which yields a big but noisy supplementary dataset $\widetilde{\mathbf{T}}_{1,\text{sup}}$. This dataset is further cleaned by a data audit module based on data Shapley scores, leading to a smaller but cleaner dataset $\mathbf{T}_{1,\text{sup}}$.

**Step 3: Learning & Deployment**. The final prediction model learns from the combination of the original task 1 data $\mathbf{T}_1$ and the supplementary data $\mathbf{T}_{1,\text{sup}}$. The resulting multi-task model $f_{\text{MTL}}$ can be used for all datasets $\mathbf{D}_* \in \mathbf{T}_1$ without any modifications. Furthermore, the model can predict for new datasets $\mathbf{D} \in \mathbf{T}$ in a zero-shot manner and perform few-shot learning for any $\mathbf{D} \in \mathbf{T}$ or $\mathbf{D} \notin \mathbf{T}$. (Note that the primary purpose of the pseudolabels is to facilitate training the zero-shot and few shot models, and are not meant to improve the performance of the original model.)

We will elaborate on the technical details of these steps in the following sections.

## 2.3 TABULAR DATA CONSOLIDATION & SANITY CHECK

The primary challenge in scaling medical tabular data predictors is the scarcity of large datasets with standardized schemas, as reflected in the sample data below.

| age | gender | height | weight | ... | mortality |
|-----|--------|--------|--------|-----|-----------|
| 18  | f      | 1.7    | 60     | ... | 0         |

| demo1 | demo2 | demo3 | ae1 | ae2 | ... | target |
|-------|-------|-------|-----|-----|-----|--------|
| 25    | 160   | 0     | 0   | 1   | ... | 14     |

Existing tabular prediction models often struggle on these datasets due to the vague semantic meanings of the varying columns and values. Our approach is to transform each row into natural language sentences that describe the sample using generative language models like GPT-3.5. Specifically, we combine the *linearization, prompting*, and *sanity check* steps to obtain input data that the language model can use to generate coherent and meaningful natural language descriptions.

**Linearization**. A function $\texttt{linearize}(\mathbf{c}, \mathbf{v})$ takes the column names $\mathbf{c}$ and the corresponding cell values $\mathbf{v}$ from a row as the input, then linearizes the row to a concatenation of paired columns and cells $\{c : v\}$. Notably, we identify sparse tabular datasets that have many binary columns. In linearization, we exclude binary columns that have a cell value of $\texttt{False}$ and only include those with positive values. This approach has two key benefits. First, it ensures that the linearized output does not exceed the input limit of language models. Second, it helps in reducing hallucinations arising from failed negation detection during the generation process of the LLMs. An ablation on the different types of tabular to text serialization is shown in Table 10 in the Appendix, showing that audited examples improve performance via augmentation. We believe that this performance benefit is useful, and serves to justify our usage of more advanced paraphrasing and auditing techniques.

**Prompting**. We combine the linearization with prefix $p$ and suffix $s$ to form the LLM prompt as $(p, \texttt{linearize}(\mathbf{c}, \mathbf{v}), s)$. The schema definition is added to $p$ to provide the context for LLM when describing the sample. For each column, we provide the type and explanation as $\texttt{\{column\}(\{type\}): \{explanation\}}$ (e.g., "*demo1(numerical): the age of the patient in years.*"). The suffix $s$ represents the instruction that steers LLM to describe the target sample or generate paraphrases for data augmentation. We describe the specific prompt templates we used in Appendix C.1 and display some consolidated examples in Appendix C.2.

The text descriptions $\mathbf{x}$ are hence sampled from LLMs by

$$\mathbf{x} \sim \text{LLM}(p, \texttt{linearize}(\mathbf{c}, \mathbf{v}), s). \tag{1}$$

The paired inputs and target $\{\mathbf{x}, y\}$ will be the training data for the tabular prediction model $f$. We can adjust the suffix $s$ in Eq. 1 to generate multiple paraphrases of the same sample as a way for instance-level data augmentation. Some instance-level augmentation examples are available in Appendix C.3.

**Sanity Check via LLM's Reflection**. To reduce low-quality generated samples, a sanity check function evaluates the fidelity of the generated text $\mathbf{x}$ to address potential hallucinations or loss of information that occurs during the translation process $\{\mathbf{c}, \mathbf{v}\} \rightarrow \mathbf{x}$ in Eq. 1, particularly for numerical features. Specifically, we query LLM with the input template "$\texttt{What is the \{column\}?}$ $\texttt{\{x\}}$" to check if the answer matches the original values in $\{\mathbf{c}, \mathbf{v}\}$. The descriptions are corrected by re-prompting the LLM if the answers do not match. We provide some examples of sanity checks in Appendix C.4, and the quantitative analysis of this correction is available in Appendix C.5.

## 2.4 DATA ENRICHMENT & REFINEMENT

Through the consolidation and sanity check process, we are able to aggregate all tabular samples $\{\mathbf{x}, y\}$ from the target task $\mathbf{T}_1$ and train a prediction model. We can use the dataset $\mathbf{T}_1$ to train a multi-task learning model, denoted as $f_{\text{MTL}}$, which can be applied to all datasets within the task. Nevertheless, there still lacks a route to leverage data from out-domain tasks $\mathbf{T}_* \sim \mathcal{T}, \forall \mathbf{T}_* \in \mathcal{T} \backslash \{\mathbf{T}_1\}$ for data enrichment. It is particularly valuable for low-data applications such as healthcare, where there may only be a few dozen data points for each dataset. Specifically, we propose to align out-domain task datasets via a *learn*, *annotate*, and *audit* pipeline for data enrichment.

**Learn and Annotate**. We train an initial model $f_{\text{MTL}}$ on all available training data from $\mathbf{T}_1$ (we will omit the subscript 1 from now on to avoid clutter). The model $f_{\text{MTL}}$ then makes pseudo labels

for a set of external samples that are retrieved from all other tasks $\mathcal{T} \setminus \{\mathbf{T}\}$, formulating the noisy supplementary data $\widetilde{\mathbf{T}}_{\text{sup}} = \{(\mathbf{x}_i, \tilde{y}_i)\}$: $\mathbf{x}_i$ are consolidated textual description samples and $\tilde{y}_i$ are noisy labels that are aligned with the objective format of the target task.

**Quality Audit**. It is vital to audit the quality of noisy training data to ensure optimal prediction performance. To this end, we clean $\widetilde{\mathbf{T}}$ by estimating the data Shapley values for each instance (Ghorbani & Zou, 2019). We denote the value function by $V(\mathbf{S})$, which indicates the performance score evaluated on the target task $\mathbf{T}$ of the predictor trained on training data $\mathbf{S}$. Correspondingly, we have the data Shapley value $\phi_i$ for any sample $(\mathbf{x}_i, \tilde{y}_i) \in \widetilde{\mathbf{T}}$ defined by

$$\phi_i = C \sum_{\mathbf{S} \subseteq \widetilde{\mathbf{T}} \setminus \{i\}} \frac{V(\mathbf{S} \bigcup \{i\}) - V(\mathbf{S})}{\binom{n-1}{|\mathbf{S}|}}, \tag{2}$$

where the summation is over all subsets of $\widetilde{\mathbf{T}}$ not with sample $i$; $n$ is the number of samples in $\widetilde{\mathbf{T}}$; $|\cdot|$ implies to the size of the set; $C$ is an arbitrary constant. Intuitively, $\phi_i$ measures the approximated expected contribution of a data point to the trained model's performance. Therefore, a sample with low $\phi$ is usually of low quality itself.

Computing the exact data Shapley value with Eq. 2 requires an exponentially large number of computations with respect to the number of train data sources. Instead, we follow (Jia et al., 2021; 2019) to use K-Nearest Neighbors Shapley (KNN-Shapley), which offers an avenue for efficient data Shapley computation. Moreover, we are able to achieve a $10\times$ speedup by parallelizing the algorithm, which completes computing scores for 100K+ samples in minutes. Upon acquiring $\Phi = \{\phi_i\}$, we execute a representative stratified sampling corresponding with the distribution of the sample classes to establish the cleaned supplementary dataset $\mathbf{T}_{\text{sup}}$. Appendix G explores the Shapley value and pseudo label distributions of different supplemental datasets.

## 2.5 LEARNING & DEPLOYMENT

After the quality check step, we obtain the original task dataset $\mathbf{T}$ and the supplementary dataset $\mathbf{T}_{\text{sup}}$ and have two potential options for model training. The first is to combine both datasets for training, but we have found that this approach results in suboptimal performance. Instead, we employ a two-step training approach: (1) pre-train the model on $\mathbf{T}_{\text{sup}}$, and (2) finetune the model using $\mathbf{T}$. The resulting model will be deployed to provide predictions for any tabular samples belonging to the target task. Because the model trained on the supplementary model has not seen any examples from the original dataset, it is able to make zero-shot predictions for the test samples from a new dataset $\mathbf{D}' \notin \mathbf{T}$ or adapt to a new task $\mathbf{T}'$ via few-shot learning when a few labeled data are available. Due to the small number of samples in some datasets, we thought it would be best to use all possible training samples in the learning phase, without leaking information in the testing phase. As the label distribution is highly skewed, it may also bias our model if validation samples were chosen randomly, and we did not have the expertise to choose a representative sample. In our case, we chose to simply train the model for 3 epochs on all of the datasets, and then perform a single pass of fine tuning, without significant hyperparameter optimization, due to the small amount of data and the good performance that it gives.

## 3 EXPERIMENTS

We conduct an extensive evaluation of `MediTab`'s performance in **supervised learning** (Q1), **few-shot learning** (Q2), and **zero-shot prediction** (Q3). We also compare the different training strategies for the final deployment of our method (Q4).

## 3.1 EXPERIMENTAL SETTING

**Datasets**: In our experiments, we introduce the following types of tabular prediction tasks. **Patient Outcome Datasets**. This dataset includes the patient records collected separately from seven oncology clinical trials [1]. These datasets each have their own unique schema and contain distinct groups

---

[1]https://data.projectdatasphere.org/projectdatasphere/html/access

Table 1: The statistics of Patient Outcome Prediction Datasets. # is short for the number of. Categorical, Binary, Numerical show the number of columns belonging to these types. N/A means no label is available for the target task. We used 20% of the data for testing.

| Trial ID | Trial Name | # Patients | Categorical | Binary | Numerical | Positive Ratio | Train/Test Split |
|---|---|---|---|---|---|---|---|
| NCT00041119 | Breast Cancer 1 | 3,871 | 5 | 8 | 2 | 0.07 | 3096 / 775 |
| NCT00174655 | Breast Cancer 2 | 994 | 3 | 31 | 15 | 0.02 | 795 / 199 |
| NCT00312208 | Breast Cancer 3 | 1,651 | 5 | 12 | 6 | 0.19 | 1320 / 331 |
| NCT00079274 | Colorectal Cancer | 2,968 | 5 | 8 | 3 | 0.12 | 2374 / 594 |
| NCT00003299 | Lung Cancer 1 | 587 | 2 | 11 | 4 | 0.94 | 469 / 118 |
| NCT00694382 | Lung Cancer 2 | 1,604 | 1 | 29 | 11 | 0.45 | 1283 / 321 |
| NCT03041311 | Lung Cancer 3 | 53 | 2 | 11 | 13 | 0.64 | 42 / 11 |
| *External Patient Database* | | | | | | | |
| MIMIC-IV | | 143,018 | 4 | 1 | 1 | N/A | |
| PMC-Patients | | 167,034 | 1 | 1 | 1 | N/A | |

Table 2: The statistics of the Clinical Trial Outcome Datasets. # is short for the number of. N/A means no label is available for the target task. We used the same data splits as Fu et al. (2022) (train and test are trials before / after 2015 respectively)

| Dataset | # Trials | # Treatments | # Conditions | # Features | Positive Ratio | Train/Test Split |
|---|---|---|---|---|---|---|
| TOP Benchmark Phase I | 1,787 | 2,020 | 1,392 | 6 | 0.56 | 1136 / 575 |
| TOP Benchmark Phase II | 6,102 | 5,610 | 2,824 | 6 | 0.50 | 4317 / 1504 |
| TOP Benchmark Phase III | 4,576 | 4,727 | 1,619 | 6 | 0.68 | 3359 / 1048 |
| *ClinicalTrials.gov Database* | | | | | | |
| Phase I-IV | 223,613 | 244,617 | 68,697 | 9 | N/A | |

of patients in different conditions. A CTGAN model (Xu et al., 2019) was trained on the raw data to generate the synthetic patient data for the experiments. We train the model to predict the patient's morbidity, which is a binary classification task. The statistics of the datasets are available in Table 1. **Clinical Trial Outcome Datasets**. We use clinical trial data from the `HINT` benchmark (Fu et al., 2022) and `ClinicalTrials.gov` [2]. The `HINT` benchmark contains drug, disease, and eligibility information for 17K clinical trials. The trial database contains 220K clinical trials with information about the trial setup (such as title, phase, enrollment, conditions, etc.). Both datasets cover phases I, II, and III trials, but only the `HINT` benchmark includes the trial outcome labels in {success, failure}. We have also included MIMIC-IV dataset and PMC-Patients dataset as the external patient database and clinical trial documents as the external trial outcome prediction dataset. Please refer to Appendix D for details.

**Implementations**: For the patient outcome prediction task, we choose a tree ensemble method (XGBoost) (Chen & Guestrin, 2016a), Multilayer Perceptron, FT-Transformer (Gorishniy et al., 2021), TransTab (Wang & Sun, 2022b), and TabLLM (Hegselmann et al., 2022) as the baselines. For the trial outcome prediction task, we choose XGBoost, feed-forward neural network (FFNN) (Tranchevent et al., 2019), DeepEnroll (Zhang et al., 2020), COMPOSE (Gao et al., 2020), HINT (Fu et al., 2022), and SPOT (Wang et al., 2023b) as the baselines. We use `PyTrial` (Wang et al., 2023a) to implement most baselines and provide the parameter tuning details of the selected baselines in Appendix E.

We use a pre-trained bidirectional transformer model named BioBERT (Lee et al., 2020) as the classifier for `MediTab`. We utilize GPT-3.5 (Brown et al., 2020) via OpenAI's API [3] for the data consolidation and enhancement. We use UnifiedQA-v2-T5 3B (Khashabi et al., 2020) [4] for the sanity check. The evaluation metrics selected are ROC-AUC and PR-AUC, with the details in Appendix F. Further ablations on base model choice is shown in Appendix H. All experiments were run with 2 RTX-3090 GPUs and AMD Ryzen 3970X 32-Core CPU.

---

[2] https://clinicaltrials.gov/

[3] Engine `gpt-3.5-turbo-0301`: https://platform.openai.com/docs/models/gpt-3-5

[4] Huggingface: allenai/unifiedqa-v2-t5-large-1363200

Table 3: Test performances on the Patient Outcome Datasets. "-" indicates not converged.

| Trial Name | Metrics | XGBoost | MLP | FT-Transformer | TransTab | TabLLM (Single Dataset) | TabLLM (Multi-Dataset) | MediTab |
|---|---|---|---|---|---|---|---|---|
| Breast Cancer 1 | AUROC | 0.5430 | 0.6091 | 0.5564 | 0.5409 | - | - | **0.6182** |
| | PRAUC | 0.0796 | 0.0963 | 0.0803 | 0.0923 | - | - | **0.1064** |
| Breast Cancer 2 | AUROC | 0.6827 | 0.6269 | 0.6231 | 0.6000 | - | - | **0.8397** |
| | PRAUC | 0.1559 | 0.1481 | 0.0520 | 0.0365 | - | - | **0.1849** |
| Breast Cancer 3 | AUROC | 0.6489 | 0.7065 | 0.6338 | 0.7100 | 0.6163 | 0.6103 | **0.7529** |
| | PRAUC | 0.3787 | 0.4000 | 0.3145 | 0.4133 | 0.3023 | 0.2977 | **0.4567** |
| Colorectal Cancer | AUROC | 0.6704 | 0.6337 | 0.5951 | 0.7096 | - | - | **0.7107** |
| | PRAUC | 0.2261 | 0.1828 | 0.1541 | 0.2374 | - | - | **0.2402** |
| Lung Cancer 1 | AUROC | - | 0.6023 | - | 0.6499 | - | - | **0.7246** |
| | PRAUC | - | 0.9555 | - | 0.9672 | - | - | **0.9707** |
| Lung Cancer 2 | AUROC | **0.6976** | 0.5933 | 0.6093 | 0.5685 | 0.6188 | 0.6279 | 0.6822 |
| | PRAUC | **0.6865** | 0.5662 | 0.5428 | 0.4922 | 0.5619 | 0.5772 | 0.6710 |
| Lung Cancer 3 | AUROC | 0.6976 | 0.6429 | 0.5357 | 0.6786 | 0.8036 | 0.6786 | **0.8928** |
| | PRAUC | 0.7679 | 0.8501 | 0.7250 | 0.7798 | 0.8256 | 0.7338 | **0.9478** |

Table 4: Test performances on the Clinical Trial Outcome Datasets.

| Trial Data | Metrics | XGBoost | FFNN | DeepEnroll | COMPOSE | HINT | SPOT | MediTab |
|---|---|---|---|---|---|---|---|---|
| Phase I | AUROC | 0.518 | 0.550 | 0.575 | 0.571 | 0.576 | 0.660 | **0.699** |
| | PRAUC | 0.513 | 0.547 | 0.568 | 0.564 | 0.567 | 0.689 | **0.726** |
| Phase II | AUROC | 0.600 | 0.611 | 0.625 | 0.628 | 0.645 | 0.630 | **0.706** |
| | PRAUC | 0.586 | 0.604 | 0.600 | 0.604 | 0.629 | 0.685 | **0.733** |
| Phase III | AUROC | 0.667 | 0.681 | 0.699 | 0.700 | 0.723 | 0.711 | **0.734** |
| | PRAUC | 0.697 | 0.747 | 0.777 | 0.782 | 0.811 | 0.856 | **0.881** |

## 3.2 RESULTS ON PATIENT OUTCOME PREDICTION AND TRIAL OUTCOME PREDICTION

We report the *supervised* results for patient outcome prediction: the AUROC and PRAUC on the test sets of all clinical trials, in Table 3. Note that we train a single classifier for MediTab and predict on all datasets, while the baselines need to be trained on each dataset separately. Our findings demonstrate that a single MediTab model achieves the highest ranking in 5 out of 7 datasets, with an overall ranking of 1.57 across all datasets. Conversely, MLP and FT-Transformer fail to converge in certain cases due to imbalanced target labels (e.g., Lung Cancer 1) or limited availability of data (e.g., Lung Cancer 3). This highlights the data-hungry nature of deep learning algorithms and emphasizes the importance of augmenting training data through data consolidation and enrichment. Additionally, we observe that TabLLM fails in both the single-dataset and multi-dataset settings. We see that with only the text-template serialization performs poorly in this setting, with multiple datasets not converging. It is possible that the small amount of data and the clinical-specific terms are too niche for the general-purpose TabLLM. Furthermore, it is not able to generalize across datasets, as the column names are quite diverse (Table 5).

MediTab also leads to substantial improvements in trial outcome prediction tasks, as illustrated in Table 4. Notably, our approach outperforms all other methods in every phase of the trials. We observe remarkable improvements of 5.9%, 9.5%, and 3.2% over the previous state-of-the-art baselines in the three phases, respectively. This provides insight into the benefits of increased data availability and the utilization of transfer learning in deep learning-based tabular prediction algorithms.

## 3.3 RESULTS ON ZERO-SHOT AND FEW-SHOT LEARNING

We assess the *zero-shot* prediction capability of MediTab on two tasks. For the evaluation of the dataset $\mathbf{D}$, we deliberately exclude $\mathbf{D}$ from the training data during step 2, where pseudo labels are generated for the external database. When computing the data Shapley values for out-domain samples during the quality check process, $\mathbf{D}$ is also excluded. Subsequently, we train a model solely on the cleaned supplementary data $\mathbf{T}_{sup}$ and evaluate its performance on the target dataset $\mathbf{D}$. The results of this evaluation are illustrated in Figure 3. MediTab exhibits impressive zero-shot performances: it wins over supervised XGBoost models in 5 out of 7 datasets in patient outcome

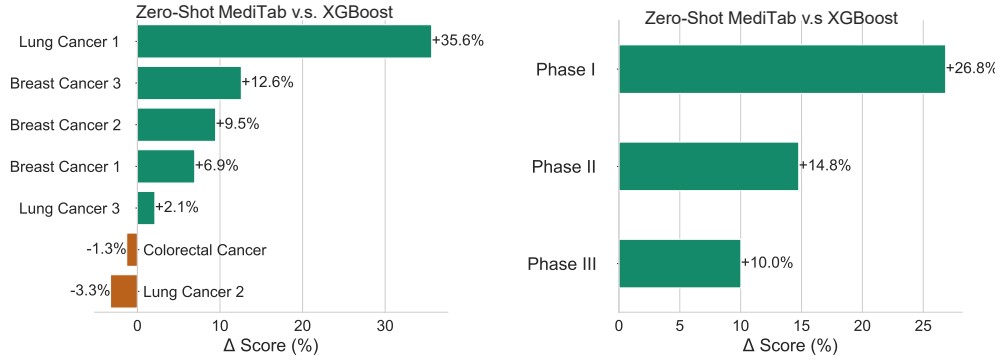

Figure 3: **Zero-shot `MediTab`** is better than a fully supervised baseline (XGBoost). The evaluation is across 7 patient outcome prediction datasets (left) and 3 trial outcome prediction datasets (right). The compared baseline XGBoost model is fitted on each dataset, respectively.

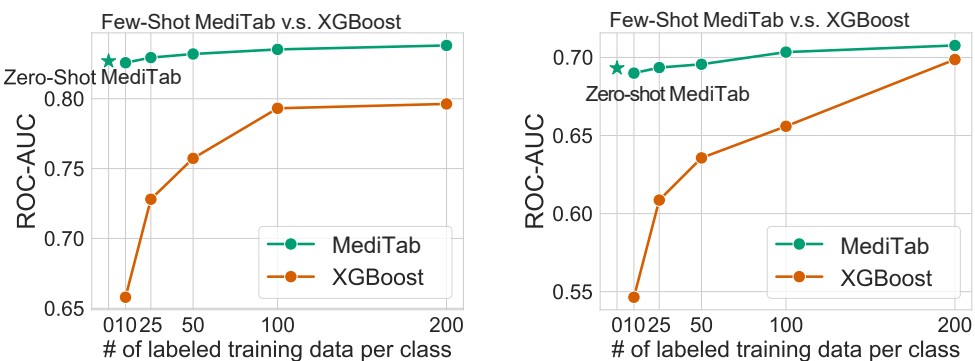

Figure 4: **Few-shot `MediTab`** compared with XGBoost with varying training data sizes. The compared baseline XGBoost model is fitted on each dataset, respectively.

prediction and all three datasets in trial outcome prediction by a significant margin. On average, `MediTab` achieves gains of 8.9% and 17.2% improvements in the two tasks, respectively.

The encouraging zero-shot learning result sheds light on the development of task-specific tabular prediction models that can offer predictions for new datasets even before the label collection stage. This becomes particularly invaluable in scenarios where acquiring training labels is costly. For instance, it enables us to predict the treatment effect of a drug on a group of patients before conducting clinical trials or collecting any trial records. Consequently, it allows us to make informed decisions regarding treatment adjustments or trial discontinuation.

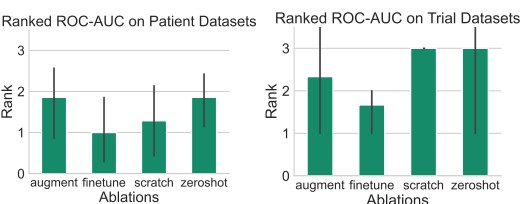

Figure 5: ROC-AUC Ranking (lower is better) of the variations of `MediTab`.

We further visualize the few-shot learning results in Figure 4. We are able to witness consistent performance improvement with more labeled training samples for both methods. Additionally, for all tested cases, XGBoost is unable to surpass the zero-shot score of `MediTab`.

### 3.4 RESULTS ON ABLATIONS ON DIFFERENT LEARNING STRATEGIES

Section 2.5 discusses a two-stage training strategy for the final learning & deployment stage. Here, we investigate the different training regimens of our method: single-stage training (`augment`), two-stage training (`finetune`), training on the original datasets from scratch (`scratch`), and zero-

shot (`zeroshot`). We list their rankings in Figure 5 and detailed performances across datasets in Tables 7 and 8. Results show that `finetune` generally performs the best. We conjecture that jointly training on the target task and supplementary data improves the model's overall utility, but it may affect the performance of specific samples in the target task **T**. Furthermore, we also identify that `zeroshot` reaches comparable performances with `scratch`.

## 4 RELATED WORK

**Tabular Prediction** has traditionally relied on tree ensemble methods (Chen & Guestrin, 2016b; Ke et al., 2017). In recent years, the powerful representation learning abilities of neural networks have motivated the new design of deep learning algorithms for tabular prediction (Arik & Pfister, 2021; Kadra et al., 2021; Chen et al., 2023; Bertsimas et al., 2022). They involve using transformer-based architectures (Huang et al., 2020; Gorishniy et al., 2021; Wang & Sun, 2022a) to enhance automatic feature interactions for better prediction performances. In addition, self-supervised learning (SSL) has been extended to tabular prediction tasks. This includes approaches such as generative pretraining objective by masked cell modeling (Yoon et al., 2020; Arik & Pfister, 2021; Nam et al., 2023), and discriminative pretraining objective by self-supervised (Ucar et al., 2021; Somepalli et al., 2022; Bahri et al., 2022) or supervised contrastive learning (Wang & Sun, 2022b). Moreover, transfer learning was also adapted to tabular prediction, employing prompt learning based on generative language models (Hegselmann et al., 2022) and multi-task learning (Levin et al., 2023). Multi-task learning and transfer learning were also performed in the medical domain for EHR-based predictive modeling (Hur et al., 2023; 2022). Nonetheless, these approaches primarily focus on *algorithm* design, including model architecture and objective functions, often overlooking the engineering of the underlying *data*.

**Data-Centric AI** underscores the importance of data for building advanced machine learning prediction systems (Zha et al., 2023). Notable progress in the domain of tabular data includes efforts to detect (Wang et al., 2020) and debug the noises in labels (Kong et al., 2021); automate feature selection (Liu et al., 2023); and streamline feature generation (Su et al., 2021). Additionally, LM for finetuning on tabular data has been proposed by (Dinh et al., 2022), but it uses strict templates to create the sentence, which limits its expressivity. These methods were proposed for general tabular data while not covering the challenges of heterogeneity and limited samples in medical tabular data. Though there were efforts in enhancing medical codes in EHRs with text descriptions (Hur et al., 2022), there is no further exploration on augmenting medical tabular data that include more diverse features. In contrast, we present a data engineering framework designed to consolidate diverse tabular datasets, by distilling the knowledge from large language models with hallucination detection and distilling from out-domain datasets with data auditing. `MediTab` is hence able to build a versatile prediction model for the target task.

## 5 CONCLUSION

In conclusion, we proposed a novel approach to train universal tabular data predictors for medical data. While there were many efforts in developing new algorithms for tabular prediction, the significance of data engineering has raised much less attention. Specifically, in medicine it is faced with challenges in limited data availability, inconsistent dataset structures, and varying prediction targets across domains. To address these challenges, `MediTab` generates large-scale training data for tabular prediction models by utilizing both in-domain tabular datasets and a set of out-domain datasets. The key component of this approach is a data engine that utilizes large language models to consolidate tabular samples by expressing them in natural language, thereby overcoming schema differences across tables. Additionally, the out-domain tabular data is aligned with the target task using a *learn, annotate, and refine* pipeline. By leveraging the expanded training data, `MediTab` can effectively work on any tabular dataset within the domain without requiring further fine-tuning, achieving significant improvements compared to supervised baselines. Moreover, `MediTab` demonstrates impressive performance even with limited examples (few-shot) or no examples (zero-shot), remaining competitive with supervised approaches across various tabular datasets.

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

**Contents**

## A    BROADER IMPACT

Tabular data, which is commonly used in various fields such as healthcare, finance, systems monitoring, and more, has received less attention compared to other domains like vision, language, and audio in terms of investigating the generalizability of its models. Non-deep algorithms, particularly tree-based methods such as XGBoost and Random Forest, still dominate the analysis of tabular data but generally be less flexible in terms of their input features and the target task. This research paper introduces `MediTab`, a framework that creates a universal tabular model in a medical domain by leveraging knowledge transfer from pre-trained LLMs.

`MediTab` excels in handling input tables with varying column numbers and leveraging publicly available unstructured text in the domain, saving significant time and resources in data engineering and cleaning. It enables knowledge transfer from models trained on sensitive data and facilitates the use of zero-shot models that match fully supervised baselines. While further research is needed to generalize `MediTab` to more application domains, it brings the promise of creating general tabular models comparable to foundational models in fields like CV and NLP.

We only use the publicly available, open-source, pre-trained models from Huggingface as our base models and baselines. This reduces the chance of data contamination and improves safety and equity regarding access to our methods. We evaluate across diverse datasets with different groups of patients to ensure it does not perpetuate bias or disparities, and all datasets are obtained, with permission, from Project Data Sphere, Github, and clinicaltrials.gov (for trial outcome prediction), where anyone can obtain the data for research purposes. The only private model is ChatGPT, but that is a current research direction we are looking into to further increase transparency and reproducibility.

Importantly, it should be noticed that we did not send any private patient records, e.g., MIMIC-IV Physionet data, to OpenAI because we do not augment these samples from the MIMIC database, similarly for PMC-patients. As shown in Table 1, there are two types of patient datasets used in the paper: (1) dataset#1: clinical trial patient data and $T$ (2) dataset#2: the external patient database. Refer to Figure 2, dataset#2 is used by MediTab in Step 2: Learn, Annotate, and Audit, to get the cleaned data $T_{sup}$ as the enrichment for the target task dataset#1, where no sample was sent to LLM for augmentation. Only samples in dataset #1 are consolidated and augmented by LLM. The model was then pre-trained on $T_{sup}$ and fine-tuned on $T$.

To avoid leaking individual patient records from dataset #1, we also did not send the raw clinical trial patient records to OpenAI but make a synthetic version of them. Technically, we used an KNN-based algorithm to generate tabular patient samples grounded on the real patient data, referring to (Beigi et al., 2022). In the future, we will develop an in-house LLaMA2 paraphraser to avoid the privacy issue. We expect the augmentation part can be replaced by local LLMs such as LLAMA2-70B after instruction tuning, but we will leave it as future work.

## B    LIMITATIONS

However, the current framework has drawbacks. Data privacy concerns arise if patient data is handled through an unsecured server vulnerable to exploitation by malicious third parties. To ensure safety, proper security procedures must be followed. If resources permit, training a private model specifically for paraphrasing would yield the safest outcome. Another issue is the tendency of language models to generate hallucinated text. While metrics exist to measure the presence of original information, measuring the creation of new, false information remains an open problem in NLP. Although some hallucinations may aid downstream models in interpreting table data, future research should include sanity checks to minimize irrelevant hallucinations. Additionally, the exploration of using external, supplemental datasets as vehicles of knowledge transfer and / or distantly supervised data via data Shapley is highly interesting, and requires more exploration. Future use cases may even reduce the risk of privacy leaks by creating datasets from publicly available, open data, that preserve performance of models trained on more sensitive data.

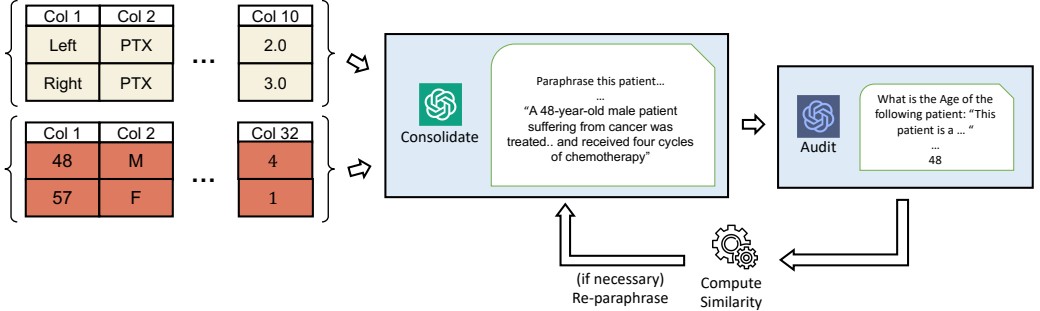

Figure 6: The paraphrasing pipeline is showing an example of a specific patient. The raw tabular data (of varying schema) is converted to unstructured natural language text by a first LLM and then audited by a secondary LLM with QA. Errors are then used to correct the initial conversion. Patient features have been changed to preserve anonymity.

## C    DATA CONSOLIDATION AND AUGMENTATION: MORE DETAILS

### C.1    TASK PROMPT TEMPLATES

In order to generate the desired text, we start by generating primitive sentences that capture the essential information from the original row of data. For categorical and numerical features, we combine the feature name and its corresponding values into a single sentence. For binary features, we include the feature name in the primitive sentence only if the value is `True` in order to avoid generating false information. We then use GPT-3.5 to paraphrase these primitive sentences. Furthermore, these texts are audited and corrected, resulting in a final text that accurately represents the original data. An illustration of this process can be found in Figure 6.

The prompt and suffix that we leverage to consolidate a row are as follows:

Listing 1: prompt for data consolidation

```
prompt = '''
Here is the schema definition of the table:

{schema_definition}

This is a sample from the table:

{linearization}

Please describe the sample using natural language.
'''
```

If we want to augment the original samples by paraphrasing, we use the prompt as follows:

Listing 2: prompt for instance-level augmentation

```
prompt = '''
Here is the schema definition of the table:

{schema_definition}

This is a sample from the table:

{linearization}

Please paraphrase the sample in 5 different ways in natural language.
'''
```

## C.2 EXAMPLE CONSOLIDATIONS

Here are 7 examples of consolidations as given after the linearization of the samples in the trial datasets (patient details changed for clarity).

The columns are all quite diverse in terms of semantic meaning, and we believe that traditional methods like data imputation or renaming / removing would not apply here, as we have fundamentally different features. Note that the features for each dataset is as follows:

1. Breast Cancer 1: race, post-menopause, human epidermal growth factor receptor 2 is positive, treatment, tumor laterality, estrogen receptor positive, progesterone receptor positive, cancer histologic grade, prior hormonal therapy, prior chemotherapy, biopsy type, sentinel node biospy, axillary dissection, number of positive axillary nodes, tumor size

2. Breast Cancer 1: age, sex, adverse effect: nausea, adverse effect: vomiting, adverse effect: asthenia, adverse effect: stomatitis, adverse effect: infection, adverse effect: pain, adverse effect: diarrhea, adverse effect: skin disorder, adverse effect: dyspnea, surgery: mastectomy, surgery: lumpectomy, surgery: quadrantectomy/segmental, multifocal tumor, histopathologic grade, histopathologic type, tumor size, number of positive axillary lymph nodes, number of resected axillary lymph nodes, estrogen receptor positive, lab test: hemoglobin, lab test: neutrophils, lab test: platelets, lab test: white blood cells, lab test: asat (sgot), lab test: alkaline phosphatase, lab test: alat (sgpt), lab test: total bilirubin, lab test: creatinine, height, weight, medical condition: history of tobacco use, medical condition: essential hypertension nos, medical condition: other bilateral destruction/occlusion of fallopian tubes, medical condition: obesity, medical condition: penicillins causing adverse effects in therapeutic use, medical condition: asthma nos, drug: dexamethasone, drug: zofran, drug: navoban, drug: kytril, drug: ondansetron, drug: tamoxifen, drug: tropisetron, drug: betapred, drug: novaban, drug: maxolon

3. Breast Cancer 3: age, adverse effect: febrile neutropenia, adverse effect: infection (documented clinically), adverse effect: infection without neutropenia(specify), adverse effect: vomiting, adverse effect: nausea, anti-tumor therapy: xeloda, anti-tumor therapy: taxotere, anti-tumor therapy: arimidex, anti-tumor therapy: zoladex, anti-tumor therapy: cyclophosphamide, number of resected axillary node, numer of positive axillary node, primary tumor size, estrogen receptor positive, progesterone receptor positive, weight, height

4. Colorectal Cancer: adherence, age, arms, serious adverse effect, bowel obstruction, bowel perforation, histology, ecog performance score, race, sex, biomarker kras, bmi, adverse effect: thrombosis, adverse effect: hypersensitivity, adverse effect: infarction, adverse effect: diarrhea

5. Lung Cancer 1: age, ecog performance status, gender, number of chemotherapy cycles, treatment arm, number of metastatic

```
       sites, weight loss larger than 10%, adverse effects:
       granulocytes/bands, adverse effects:  wbc, adverse effects:
       lymphocytes, adverse effects:  other miscellaneous 1,
       adverse effects:  platelets, adverse effects:  hemoglobin,
       adverse effects:  dyspnea, adverse effects:  nausea, adverse
       effects:  other miscellaneous 2, adverse effects:  other
       neurologic
```

6. Lung Cancer 2:  historical disease:  deep vein thrombosis, historical disease:  pulmonary embolism, historical disease: antiandrogen therapy, historical disease:  cardiac failure chronic, historical disease:  chronic respiratory failure, historical disease:  venous insufficiency, historical disease:  coronary artery disease, historical disease: myocardial infarction, historical disease:  hypertension, historical disease:  peripheral arterial occlusive disease, medication:  dexamethasone, medication:  ondansetron, medication:  heparin, medication:  fluorouracil, medication:  ranitidine, medication:  cisplatin, medication: metoclopramide, medication:  carboplatin, medication: furosemide, aderse effect:  severe malignant neoplasm progression, aderse effect:  severe neutropenia, aderse effect:  severe thrombocytopenia, aderse effect:  severe anaemia, aderse effect:  severe vomiting, aderse effect: severe pneumonia, aderse effect:  severe diarrhoea, aderse effect:  severe abdominal pain, aderse effect: severe sepsis, aderse effect:  severe leukopenia, age, sex, lab test:  hemoglobin, lab test:  leukocytes, lab test:  creatinine clearance, lab test:  creatinine, lab test:  platelet, lab test:  bilirubin, lab test:  alanine aminotransferase, lab test:  aspartate aminotransferase, lab test:  neutrophils, lab test:  alkaline phosphatase

7. Lung Cancer 3:  age, ecog performance score, sex, height, weight, brain metastasis, lactate dehydrogenase isoenzymes test abnormal, smoke status, adverse effect:  neutropenia, adverse effect:  anaemia, adverse effect:  neutrophil count decreased, adverse effect:  thrombocytopenia, adverse effect:  platelet count decreased, drug: dexamethasone, drug:  ondansetron, drug:  prednisone, drug:  sodium chloride, lab test:  leukocytes (10$\hat{9}$/l), lab test:  hemoglobin (g/l), lab test:  platelets (10$\hat{9}$/l), lab test:  lymphocytes (10$\hat{9}$/l), lab test: platelets/lymphocytes, lab test:  neutrophils (10$\hat{9}$/l), lab test:  neutrophils/lymphocytes, lab test:  monocytes (10$\hat{9}$/l), lab test:  eosinophils (10$\hat{9}$/l)

Note that the only variables that occur more than once are:

1. 6 occurrences: age

2. 4 occurrences: sex

3. 3 occurrences: weight, estrogen receptor positive, height

4. 2 occurrences: lab test: alkaline phosphatase, lab test: hemoglobin, lab test: neutrophils, adverse effect: nausea, lab test: creatinine, drug: ondansetron, tumor size, progesterone receptor positive, adverse effect: diarrhea, drug: dexamethasone, race, ecog performance score, adverse effect: vomiting

**Breast Cancer 1**

Table 5: Table of number of variables shared between datasets. # Vars Shared denotes if a variable name occurs in AT LEAST one other dataset.

| Dataset Name | # Vars Shared | # Vars Total |
|---|---|---|
| Breast Cancer 1 | 4 | 15 |
| Breast Cancer 2 | 15 | 48 |
| Breast Cancer 3 | 7 | 18 |
| Colorectal Cancer | 5 | 16 |
| Lung Cancer 1 | 1 | 17 |
| Lung Cancer 2 | 6 | 41 |
| Lung Cancer 3 | 7 | 26 |

```
race White; treatment paclitaxel;
tumor laterality left; cancer
histologic grade Low; biopsy
type incisional; post-menopause
; estrogen receptor positive ;
progesterone receptor positive ;
prior hormonal therapy ; number
of positive axillary nodes 0;
tumor size 3.0
```

```
The patient has been confirmed to
be positive for human epidermal
growth factor receptor 2.  She is
of White race and is currently
undergoing treatment using
paclitaxel.  The tumor is on the
patient's left side and has a
low histologic grade.  The biopsy
type used was incisional, and
the patient is post-menopausal.
Additionally, the patient is
estrogen receptor positive,
progesterone receptor positive,
and has undergone prior hormonal
therapy.  The tumor size is
approximately 3.0 units.
```

**Breast Cancer 2**

sex Female; histopathologic grade Moderately differentiated; histopathologic type Infiltrating ductal carcinoma; adverse effect: diarrhea ; surgery:  mastectomy ; multifocal tumor ; estrogen receptor positive ; medical condition:  history of tobacco use ; drug:  zofran ; drug: tamoxifen ; age 48.0; tumor size 2.5; number of positive axillary lymph nodes 3.0; number of resected axillary lymph nodes 19.0; lab test:  hemoglobin 7.59; lab test:  neutrophils 3.03; lab test:  platelets 327.65; lab test:  white blood cells 3.78; lab test:  asat (sgot) 30.846; lab test:  alkaline phosphatase 172.385; lab test:  alat (sgpt) 38.0; lab test:  total bilirubin 7.462; lab test:  creatinine 64.9; height 166.0; weight 84.6

The patient, a 48-year-old female, had moderately differentiated infiltrating ductal carcinoma and underwent lumpectomy and mastectomy surgeries for a multifocal tumor.  She has a medical history of tobacco use and experienced adverse effects from penicillins.  Her lab tests indicate a hemoglobin of 7.59, slightly low neutrophils at 3.03, high platelets at 327.65, and a low white blood cell count of 3.78.  She also has elevated asat (sgot) levels at 30.846, slightly high alkaline phosphatase at 172.385, alat (sgpt) at 38.0, total bilirubin at 7.462, and a creatinine of 64.9.  She also suffered from vomiting and diarrhea, which were side effects of her medication, though she was given zofran to alleviate those symptoms.

**Breast Cancer 3**

estrogen receptor positive ; progesterone receptor positive ; age 64.0; number of resected axillary node 11.0; numer of positive axillary node 7.0; primary tumor size 3.0; weight 113.0; height 162.0

The patient, a 64-year-old with estrogen and progesterone receptor positive breast cancer, underwent surgery to remove 11 axillary nodes, with 7 testing positive.  She was prescribed anti-tumor therapies including Xeloda, Taxotere, Arimidex, Zoladex, and Cyclophosphamide. Unfortunately, she experienced the adverse effect of febrile neutropenia.

**Colorectal Cancer**

arms Oxaliplatin + 5-fluorouracil/Leucovorin + Cetuximab; histology well differentiated; race white; sex female; biomarker kras wild-type; adherence ; age 45.0; ecog performance score 1; bmi 24.883

The patient experienced a negative reaction of blood clotting, specifically thrombosis, as well as a harmful effect called infarction in their arms as a result of taking Oxaliplatin + 5-fluorouracil/Leucovorin + Cetuximab.

**Lung Cancer 1**

```
gender male; treatment arm
paclitaxel, cisplatin,
etoposide,G-CDF; adverse effects:
granulocytes/bands ; adverse
effects:  wbc ; adverse effects:
lymphocytes ; adverse effects:
platelets ; adverse effects:
dyspnea ; adverse effects:
nausea ; adverse effects:  other
miscellaneous 2 ; age 76.0;
number of metastatic sites 1.0;
number of chemotherapy cycles
2.0; ecog performance status 0.0
```

```
A 76-year-old man receiving
paclitaxel, cisplatin, etoposide,
G-CDF for his single metastatic
site experienced a variety of
adverse effects, including
dyspnea, nausea, and other
miscellaneous symptoms.  He
also had decreased levels of
granulocytes/bands, white
blood cells, lymphocytes, and
platelets.
```

**Lung Cancer 2**

```
sex M; aderse effect:  severe
malignant neoplasm progression
; aderse effect:  severe
neutropenia ; aderse effect:
severe thrombocytopenia ; aderse
effect:  severe anaemia ; aderse
effect:  severe vomiting ; aderse
effect:  severe pneumonia ;
aderse effect:  severe diarrhoea
; aderse effect:  severe
abdominal pain ; aderse effect:
severe sepsis ; aderse effect:
severe leukopenia ; medication:
dexamethasone ; medication:
ondansetron ; medication:
heparin ; medication:  ranitidine
; medication:  cisplatin ;
medication:  metoclopramide
; medication:  carboplatin ;
medication:  furosemide ; age
67; lab test:  hemoglobin 134.0;
lab test:  leukocytes 4.3; lab
test:  creatinine clearance
92.489; lab test:  creatinine
84.0; lab test:  platelet 256.0;
lab test:  bilirubin 18.5; lab
test:  alanine aminotransferase
36.1; lab test:  aspartate
aminotransferase 17.9; lab test:
neutrophils 2.58; lab test:
alkaline phosphatase 556.0
```

```
The patient, a 67-year-old
male, has undergone lab tests
revealing that their hemoglobin
is 134.0, leukocytes are 4.3,
creatinine clearance is 92.489,
creatinine is 84.0, platelet
count is 256.0, bilirubin is
18.5, alanine aminotransferase is
36.1, aspartate aminotransferase
is 17.9, neutrophils are 2.58,
and alkaline phosphatase is
556.0.  They have a medical
history of antiandrogen therapy,
chronic cardiac failure, and
venous insufficiency.  However,
the patient has experienced
adverse effects, including severe
malignant neoplasm progression,
neutropenia, thrombocytopenia,
anemia, vomiting, pneumonia,
diarrhea, abdominal pain,
sepsis, and leukopenia, due to
the medications dexamethasone,
ondansetron, heparin, ranitidine,
cisplatin, metoclopramide,
carboplatin, and furosemide.
```

**Lung Cancer 3**

```
sex M; smoke status Former
Smoker; lactate dehydrogenase
isoenzymes test abnormal ;
adverse effect:  neutropenia
; adverse effect:  anaemia ;
adverse effect:  thrombocytopenia
; drug:  ondansetron ; drug:
prednisone ; age 0.671; ecog
performance score 1; height
0.805; weight 0.517; lab test:
leukocytes (10^9/l) 4.996;
lab test:  hemoglobin (g/l)
108.816; lab test:  platelets
(10^9/l) 175.895; lab test:
lymphocytes (10^9/l) 1.039; lab
test:  platelets/lymphocytes
176.287; lab test:  neutrophils
(10^9/l) 3.367; lab test:
neutrophils/lymphocytes 3.221;
lab test:  monocytes (10^9/l)
0.509; lab test:  eosinophils
(10^9/l) 0.068
```

```
The patient's lab results show a
leukocyte count of 4.996 x 10^9/L,
hemoglobin level of 108.816
g/L, platelet count of 175.895
x 10^9/L, platelet/lymphocyte
ratio of 176.287, neutrophil
count of 3.367 x 10^9/L,
neutrophil/lymphocyte ratio of
3.221, and eosinophil count of
0.068 x 10^9/L. The patient has
experienced adverse effects
of decreased neutrophil count,
platelet count, neutropenia,
anemia, and thrombocytopenia.
They were prescribed ondansetron
and prednisone.  The patient is
male, a former smoker, and has
an ecog performance score of 1.
They are 0.671 years old and have
a height of 0.805 and weight of
0.517.
```

## C.3 DATA AUGMENTATION

We show a couple of examples of data augmentation as obtained by paraphrasing 5 times:

```
1.  The patient is White, and receiving paclitaxel for a
left-sided tumor with Low histologic grade.  An incisional biopsy
was performed on the tumor.  The patient is post-menopausal and
has estrogen and progesterone receptor-positive cancer.  They had
prior hormonal therapy and no positive axillary nodes.  The tumor
size is 3.0.
2.  A left-sided tumor with Low histologic grade is being treated
with paclitaxel for a White patient who had an incisional biopsy.
The patient is post-menopausal and has estrogen and progesterone
receptor-positive cancer, as well as prior hormonal therapy.
There are zero positive axillary nodes and the tumor size is 3.0.
3.  For a White patient, paclitaxel is the treatment for a
left-sided tumor with Low histologic grade.  An incisional biopsy
was performed, and the patient is post-menopausal with estrogen
and progesterone receptor-positive cancer.  They had prior
hormonal therapy, zero positive axillary nodes, and the tumor size
is 3.0.
4.  The tumor laterality is left for a White patient receiving
paclitaxel, with a Low histologic grade.  The biopsy type was
incisional, and the patient is post-menopausal with estrogen and
progesterone receptor-positive cancer.  They had prior hormonal
therapy and no positive axillary nodes, with a tumor size of 3.0.
5.  An incisional biopsy was performed on a left-sided, Low
histologic grade tumor for a White patient receiving paclitaxel.
They are post-menopausal with estrogen and progesterone
receptor-positive cancer, having had prior hormonal therapy.
There are zero positive axillary nodes, and the tumor size is 3.0.
```

```
1.  The patient is a 62-year-old white woman with
well-differentiated histology.  She is taking arms Oxaliplatin,
5-fluorouracil/Leucovorin, and Cetuximab.  She has Kras wild-type
biomarkers and an ECOG performance score of 1.  Her BMI is 24.883
and she is adhering to her treatment plan.
2.  A white female patient with well-differentiated histology
is being treated with a combination of Oxaliplatin,
5-fluorouracil/Leucovorin, and Cetuximab.  She is 62 years old,
has Kras wild-type biomarkers, and an Ecog performance score of 1.
She is maintaining a BMI of 24.883 and adhering to her medication.
3.  An adherence patient, who is a 62-year-old white woman, has
well-differentiated histology and is taking arms Oxaliplatin,
5-fluorouracil/Leucovorin, and Cetuximab.  Her Kras biomarkers
are wild-type and she has an Ecog performance score of 1.  She is
maintaining a BMI of 24.883.
4.  The patient is a well-differentiated white female with
Kras wild-type biomarkers.  She is being treated with arms
Oxaliplatin, 5-fluorouracil/Leucovorin, and Cetuximab and has an
ECOG performance score of 1.  She is 62 years old, adhering to her
medication, and has a BMI of 24.883.
5.  A 62-year-old white woman with good histology is taking arms
Oxaliplatin, 5-fluorouracil/Leucovorin, and Cetuximab.  She has
Kras wild-type biomarkers, an Ecog performance score of 1, and is
adhering to her medication.  Her BMI is 24.883.
```

## C.4 SANITY CHECK WITH LLM'S RELECTION

The generated text must undergo auditing to ensure practical suitability for downstream tasks and human interpretability. This is especially critical when dealing with tabular data, as accurate paraphrasing is essential for maintaining important information that can significantly impact model performance. Thus, verifying faithful paraphrasing of the data is of utmost importance.

To evaluate the fidelity of the paraphrased text, we employ a cutting-edge Question-Answering model to test the paraphrased text. For different features, we query the model using the prompts below.

Listing 3: prompt for sanity check of categorical features

```
1 prompt = '''
2 {consolidated_output}
3
4 What is the value of {feature_name}?
5 '''
```

Listing 4: prompt for sanity check of binary features

```
1 prompt = '''
2 {consolidated_output}
3
4 Is {feature_name} present in the above paragraph? (a) yes (b) no.
5 '''
```

We then compare the QA model's answers with the actual feature values and calculate a normalized edit distance score (edit distance of longest string) as a measure of the paraphrase's accuracy:

$$\hat{D} = 1 - \frac{D}{\max(l_1, l_2)}, \tag{3}$$

where $D$ represents the edit distance of two strings; $l_1$ and $l_2$ are length of two input strings.

If the matching metric is too low, we add it to the list of features we wish to preserve. Then, we rerun the paraphrasing with a secondary prompt as shown here:

Listing 5: prompt for correcting a generated text

```
1  prompt = '''
2  Here is the schema definition of the table:
3
4  {schema_definition}
5
6  Please paraphrase the following in natural language.
7
8  {previous_paraphrase} + {linearization_of_missed_features}
9  '''
```

### C.4.1 EXAMPLES

In this case, the first paraphrase completely left out the severe adverse effects.

```
A male patient, aged 70.0
years, whose race is white
having poorly differentiated
histology and a wild-type Kras
biomarker, is taking Oxaliplatin
and 5-fluorouracil/Leucovorin
along with Cetuximab.  His ECOG
performance score is 0, and BMI
is 27.554.
```

```
A male patient who is white
and 70 years old is currently
being treated with Oxaliplatin
+ 5-fluorouracil/Leucovorin
+ Cetuximab for poorly
differentiated histology with
kras wild-type biomarkers.  He
has a BMI of 27.554 and a ECOG
performance score of 0.  There
have been severe adverse effects
of thrombosis, hypersensitivity,
infarction, and diarrhea,
and there is a risk of bowel
perforation.
```

In this case, the first paraphrase did not preserve the lab results:

```
This male patient has a history
of smoking but is now a former
smoker.  Unfortunately, he has
developed brain metastasis and an
abnormal lactate dehydrogenase
isoenzymes test result.  He
experienced neutropenia as
a side effect of treatment.
Additionally, he has a relatively
young age, high ecog performance
score, and low height and weight.
His lab tests show elevated
leukocytes and platelets, low
lymphocytes and hemoglobin, and
abnormal neutrophils, monocytes,
and eosinophils.
```

```
The male patient, who is a
former smoker, has a brain
metastasis and abnormal lactate
dehydrogenase isoenzymes test
result.  He is 0.429 years old
and has an ecog performance score
of 2.  With a height of 0.732
and weight of 0.517, his lab
tests reveal a leukocyte count
of 8.371 (10^9/l), hemoglobin
level of 109.0 (g/l), platelet
count of 337.286 (10^9/l),
lymphocyte level of 0.917
(10^9/l), platelet to lymphocyte
ratio of 815.19, neutrophil
count of 6.923 (10^9/l),
neutrophil to lymphocyte ratio
of 6.187, monocyte count of
0.424 (10^9/l), and eosinophil
count of 0.039 (10^9/l).  The
patient experienced adverse
effects of anaemia, decreased
neutrophil and platelet count,
and neutropenia, while being
treated with dexamethasone,
ondansetron, prednisone, and
sodium chloride.
```

## C.5 EVALUATION OF HALLUCINATIONS PRODUCED IN CONSOLIDATION

Table 6: The Mean Normalized Edit Distance (MNED). Higher is better, the range is [0,1]. For each dataset as calculated on the feature values retrieved by the QA model vs the actual feature value.

| Dataset | MNED | MNED (After Correction) |
|---|---|---|
| Breast Cancer 1 | 0.7197 | 0.8421 |
| Breast Cancer 2 | 0.5007 | 0.6578 |
| Breast Cancer 3 | 0.3463 | 0.7155 |
| Colorectal Cancer | 0.5498 | 0.7244 |
| Lung Cancer 1 | 0.4287 | 0.6326 |
| Lung Cancer 2 | 0.5059 | 0.6456 |
| Lung Cancer 3 | 0.3216 | 0.4218 |

We show the quantitative analysis of the hallucinations during the sanity check in Table 6. We see that the normalized Edit Distance (Eq. 3) shows that the paraphrasing may not preserve all of the original values of the row, but that also, re-paraphrasing does significantly help. However, this is not as bad as of an issue as it may appear. Since the performance with data augmentation is higher than without. Mean Normalized Edit Distance only measures the character differences between the strings. Upon manual inspection, there are many cases where feature values are paraphrased into a similar meaning. In other cases, the GPT-3.5 model does fail to summarize the patient completely. For example, in the Lung Cancer datasets, there are many lab numerical values that the generator fails to preserve within a reasonable max length of generated text, even with multiple tries. Further work is required to generate better paraphrases as well as improved metrics.

## D EXTERNAL DATABASE

**MIMIC-IV.** We downloaded the raw MIMIC-IV dataset (Johnson et al., 2023) and preprocessed it with a standard opensource data extraction pipeline[5]. We further extracted the patients' age, gender, diagnosis, medications, procedures, and patient notes to build the EHR patient database as the external data for `MediTab`.

**PMC-Patients.** We used the PMC-Patient dataset (Zhao et al., 2022) where 167K patient notes from 141K PubMed articles are provided. Here, we involved the patient's age, gender, and the patient notes as the features.

**ClinicalTrials.Gov.** We obtained a dump from the clinicaltrials.gov database to get all the available clinical trial documents. We used the trial's title, study type, number of enrollment, phase, conditions, interventions, and eligibility criteria, as the features to predict the trial outcome.

## E BASELINE MODELS

The baselines for **patient outcome prediction** datasets:

- XGBoost (Chen & Guestrin, 2016b): This is a tree ensemble method augmented by gradient-boosting. We use its official implementation of Python interface [6] in our experiments. We use ordinal encoding for categorical and binary features and standardize numerical features via `scikit-learn` (Pedregosa et al., 2011). We encode textual features, e.g., patient notes, via a pre-trained BioBERT (Lee et al., 2020) model. The encoded embeddings are fed to XGBoost as the input. We tune the model using the hyperparameters: $max\_depth$ in $\{4, 6, 8\}$; $n\_estimator$ in $\{50, 100, 200\}$; $learning\_rate$ in $\{0.1, 0.2\}$; We take early-stopping with patience of 5 rounds.
- Multilayer Perceptron (MLP): This is a simple neural network built with multiple fully-connected layers. We use the implementation from the individual outcome prediction module of `PyTrial`

---

[5]MIMIC-IV-Data-Pipeline: https://github.com/healthylaife/MIMIC-IV-Data-Pipeline

[6]DMLC XGBoost: https://xgboost.readthedocs.io/en/stable/

[7]. The model is with 2 dense layers where each layer has 128 hidden units. We tune the model using the hyperparameters: *learning_rate* in {1e-4,5e-4,1e-3}; *batch_size* in {32, 64}; We take the max training *epochs* of 10; *weight_decay* of 1e-4.

- FT-Transformer (Gorishniy et al., 2021): This is a transformer-based tabular prediction model. We use the implementation from the individual outcome prediction module of PyTrial. The model is with 2 transformer modules where each module has 128 hidden units in the attention layer and 256 hidden units in the feed-forward layer. We use multi-head attention with 8 heads. We tune the model using the hyperparameters: *learning_rate* in {1e-4,5e-4,1e-3}; *batch_size* in {32, 64}; We take the max training *epochs* of 10 and *weight_decay* of 1e-4.

- TransTab (Wang & Sun, 2022b): This is a transformer-based tabular prediction model that is able to learn from multiple tabular datasets. Following the transfer learning setup of this method, we take a two-stage training strategy: first, train it on all datasets in the task, then fine-tune it on each dataset and report the evaluation performances. We use the implementation from the individual outcome prediction module of PyTrial. The model is with 2 transformer modules where each module has 128 hidden units in the attention layer and 256 hidden units in the feed-forward layer. We use multi-head attention with 8 heads. We tune the model using the hyperparameters: *learning_rate* in {1e-4,5e-4,1e-3}; *batch_size* in {32, 64}; We take the max training *epochs* of 10 and *weight_decay* of 1e-4.

- TabLLM (Hegselmann et al., 2022): This is the most similar baseline to our model, as it also converts tabular data into natural paraphrases on which text classification is then performed. However, this model has some major differences from our work. First, they show that using GPT to paraphrase the tabular data performs the best, but do not perform any data-auditing to ensure that all of the information is preserved. Additionally, they do not augment their datasets. Finally, since many of their tasks general tabular classification datasets, they are not able to take full advantage of pretraining on multiple similar dataset (which we have shown performs better than training from scrath in Table 7 and Table 8). We use the default training parameters as for the baselines in the official github from https://github.com/clinicalml/TabLLM, and finetune for steps in {25K, 50K, 100K}. We primarily experiment with number of steps as that is the main difference in their finetuning parameter choice in their training script.

The baselines for **clinical trial outcome prediction** datasets:

- XGBoost (Chen & Guestrin, 2016b): This is a tree ensemble method augmented by gradient-boosting. We follow the setup used in (Fu et al., 2022).

- FFNN (Tranchevent et al., 2019): It is a feed-forward neural network, which has 3 fully-connected layers with the dimensions of dim-of-input-feature, 500, and 100, and ReLU activations. We follow the setup used in (Fu et al., 2022).

- DeepEnroll (Zhang et al., 2020): It was originally aimed at facilitating patient-trial matching, encompassing a hierarchical embedding network designed to encode disease ontology. Additionally, an alignment model was incorporated to capture the interconnections between eligibility criteria and disease information. We follow the setup used in (Fu et al., 2022).

- COMPOSE (Gao et al., 2020): Initially, its application revolved around patient-trial matching, employing a combination of a convolutional neural network and a memory network. The convolutional neural network encoded eligibility criteria, while the memory network encoded diseases. To enhance trial outcome prediction, the model's embedding was concatenated with a molecule embedding using MPNN (Message Passing Neural Network). We follow the setup used in (Fu et al., 2022).

- HINT (Fu et al., 2022): It integrates several key components. Firstly, there is a drug molecule encoder utilizing MPNN (Message Passing Neural Network). Secondly, a disease ontology encoder based on GRAM is incorporated. Thirdly, a trial eligibility criteria encoder leveraging BERT is utilized. Additionally, there is a drug molecule pharmacokinetic encoder, and a graph neural network is employed to capture feature interactions. Subsequently, the model feeds the interacted embeddings into a prediction model for accurate outcome predictions. We follow the setup used in (Fu et al., 2022).

---

[7]pytrial.indiv_outcome: https://pytrial.readthedocs.io/en/latest/pytrial.tasks.indiv_outcome.html

- SPOT (Wang et al., 2023b): The Sequential Predictive Modeling of Clinical Trial Outcome (SPOT) is an innovative approach that follows a sequential process. Initially, it identifies trial topics to cluster the diverse trial data from multiple sources into relevant trial topics. Next, it generates trial embeddings and organizes them based on topic and timestamp, creating structured clinical trial sequences. Treating each trial sequence as an individual task, SPOT employs a meta-learning strategy, enabling the model to adapt to new tasks with minimal updates swiftly. We follow the setup used in (Wang et al., 2023b).

## F    EVALUATION METRICS

We consider the following performance metrics: (1) **AUROC**: the area under the Receiver Operating Characteristic curve summarizes the trade-off between the true positive rate (TPR) and the false positive rate (FPR) with the varying threshold of FPR. In theory, it is equivalent to calculating the ranking quality by the model predictions to identify the true positive samples. However, better AUROC does not necessarily indicate better outputting of well-calibrated probability predictions. (2) **PRAUC**: the area under the Precision-Recall curve summarizes the trade-off between the precision (PPV) and the recall (TPR) with the varying threshold of recall. It is equivalent to the average of precision scores calculated for each recall threshold and is more sensitive to the detection quality of true positives from the data, e.g., identifying which trial is going to succeed.

## G    DATA SHAPLEY DISTRIBUTIONS

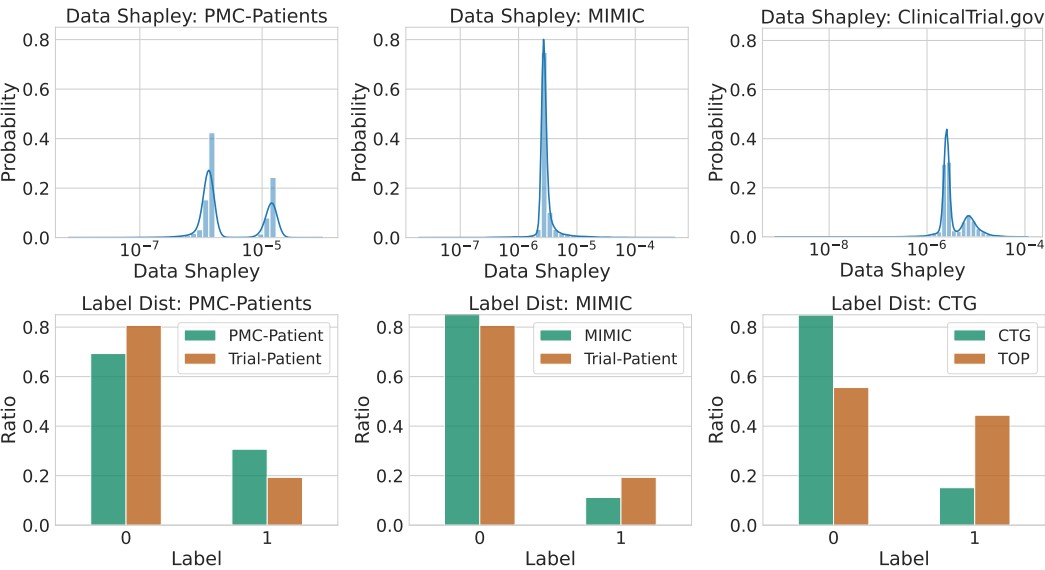

Figure 7: A plot of the distributions of the data Shapley values as well as the label ratio of 3 datasets. Note that to simplify the setup, we use the simple baseline features of the patients and their clinical discharge notes as the external datasets to obtain shapley values and pseudolabels. We see that in both the PMC-Patients and ClinicalTrial.gov datasets, there exist a bimodal distribution of shapley values, indicating that there exists a subset of data subset that is more relevant than the rest. The MIMIC distribution is more unimodal, albeit it is slightly right tailed. We also see that the predicted labels generally match the original distribution's labels, which indicates that the generated supplemental dataset does indeed preserve the true label imbalance.

# H  ABLATION RESULTS

Table 7: Ablations of `MediTab` with its variants (Retrained). The evaluation is made on patient outcome prediction datasets.

| Trial Name | Metrics | Augment | Finetune | Scratch | Zero-Shot |
|---|---|---|---|---|---|
| Breast Cancer 1 | ROC-AUC | **0.624** | 0.617 | 0.591 | 0.591 |
| | PR-AUC | **0.111** | 0.103 | 0.094 | 0.102 |
| Breast Cancer 2 | ROC-AUC | 0.713 | **0.841** | 0.803 | 0.742 |
| | PR-AUC | 0.049 | **0.071** | 0.060 | 0.051 |
| Breast Cancer 3 | ROC-AUC | 0.734 | **0.741** | 0.721 | 0.731 |
| | PR-AUC | 0.456 | **0.486** | 0.437 | 0.391 |
| Colorectal Cancer | ROC-AUC | 0.677 | 0.697 | **0.705** | 0.662 |
| | PR-AUC | 0.236 | 0.244 | **0.267** | 0.207 |
| Lung Cancer 1 | ROC-AUC | 0.623 | **0.822** | 0.699 | 0.678 |
| | PR-AUC | 0.963 | **0.987** | 0.971 | 0.969 |
| Lung Cancer 2 | ROC-AUC | 0.682 | 0.677 | **0.711** | 0.677 |
| | PR-AUC | 0.673 | 0.669 | **0.691** | 0.671 |
| Lung Cancer 3 | ROC-AUC | **0.893** | **0.893** | **0.893** | **0.893** |
| | PR-AUC | 0.948 | 0.948 | **0.957** | 0.938 |

Table 8: Ablations of `MediTab` with its variants (Retrained). The evaluation is made on clinical trial outcome prediction datasets.

| Trial Data | Metrics | Augment | Finetune | Scratch | Zero-Shot |
|---|---|---|---|---|---|
| Phase I | ROC-AUC | **0.706** | 0.701 | 0.699 | 0.657 |
| | PR-AUC | 0.725 | 0.722 | **0.726** | 0.704 |
| Phase II | ROC-AUC | 0.718 | **0.726** | 0.706 | 0.689 |
| | PR-AUC | **0.747** | 0.743 | 0.733 | 0.728 |
| Phase III | ROC-AUC | 0.724 | 0.729 | 0.726 | **0.734** |
| | PR-AUC | 0.863 | 0.876 | **0.881** | 0.877 |

Table 9: Ablations of different base models (BERT (Devlin et al., 2018), BioBERT (Lee et al., 2020), ClinicalBERT (Alsentzer et al., 2019), and TabLLM (Hegselmann et al., 2022)) in terms of downstream performance, trained from scratch respectively. The evaluation is made on clinical trial outcome prediction datasets. We see that the model selection choice is similar for BioBERT and Clinical BERT. TabLLM does not converges for 4 out of the 7 datasets. We believe that this may be due to the small amount of training data or the domain-specificity, but further research should be done to fully investigate this behavior.

| Trial Name | Metrics | BERT | ClinicalBERT | BioBERT | TabLLM |
|---|---|---|---|---|---|
| Breast Cancer 1 | ROC-AUC | 0.588 | 0.581 | 0.591 | - |
| | PR-AUC | 0.097 | 0.082 | 0.094 | - |
| Breast Cancer 2 | ROC-AUC | 0.485 | 0.724 | 0.803 | - |
| | PR-AUC | 0.023 | 0.026 | 0.060 | - |
| Breast Cancer 3 | ROC-AUC | 0.696 | 0.734 | 0.721 | 0.616 |
| | PR-AUC | 0.392 | 0.366 | 0.437 | 0.302 |
| Colorectal Cancer | ROC-AUC | 0.613 | 0.700 | 0.705 | - |
| | PR-AUC | 0.233 | 0.186 | 0.267 | - |
| Lung Cancer 1 | ROC-AUC | 0.555 | 0.479 | 0.699 | - |
| | PR-AUC | 0.962 | 0.949 | 0.971 | - |
| Lung Cancer 2 | ROC-AUC | 0.544 | 0.616 | 0.711 | 0.619 |
| | PR-AUC | 0.483 | 0.616 | 0.691 | 0.562 |
| Lung Cancer 3 | ROC-AUC | 0.357 | 0.893 | 0.893 | 0.804 |
| | PR-AUCAUC | 0.695 | 0.957 | 0.957 | 0.826 |

Table 10: Ablations of pretraining `MediTab` on different types of serialization. Simple Text refers to a simple table-to-text like "column name: column value". We see that while this simple serialization works well, we also discovered that augmenting with additional paraphrased examples indeed improves performance. Furthermore, we find that audited examples improve performance the most. Our results shown in the basic Simple Text approach, similar to what's demonstrated in TabLLM, was effective. However, we also observed that enhancing this format with paraphrased examples led to better performance. Furthermore, we find that audited examples improve performance the most. We believe that this performance benefit is useful and serves to justify our usage of more advanced paraphrasing and auditing techniques.

| Trial Name | Metric | Simple Text | Praphrase | Audited Paraphrase |
|---|---|---|---|---|
| Breast Cancer 1 | ROC-AUC | 0.607 | 0.620 | 0.617 |
| | PR-AUC | 0.098 | 0.107 | 0.105 |
| Breast Cancer 2 | ROC-AUC | 0.753 | 0.753 | 0.876 |
| | PR-AUC | 0.083 | 0.083 | 0.135 |
| Breast Cancer 3 | ROC-AUC | 0.760 | 0.758 | 0.764 |
| | PR-AUC | 0.452 | 0.481 | 0.476 |
| Colorectal Cancer | ROC-AUC | 0.695 | 0.691 | 0.705 |
| | PR-AUC | 0.259 | 0.264 | 0.256 |
| Lung Cancer 1 | ROC-AUC | 0.699 | 0.737 | 0.717 |
| | PR-AUC | 0.975 | 0.979 | 0.972 |
| Lung Cancer 2 | ROC-AUC | 0.699 | 0.697 | 0.716 |
| | PRAUC | 0.679 | 0.680 | 0.715 |
| Lung Cancer 3 | ROC-AUC | 0.607 | 0.893 | 0.929 |
| | PR-AUC | 0.697 | 0.957 | 0.968 |

Table 11: Zero-shot performance of Different Shapley Value cutoffs percentiles. After each percentile was calculated, the full retraining was performed to obtain the ROC-AUC and PR-AUC. We see that empirically, using the 50 percentile cutoff performs the best. A small cutoff allows in too many irrelevant examples, and a high cutoff may remove too much diversity from the data.

| Trial Name | Metric | 10% | 25% | 50% | 90% |
|---|---|---|---|---|---|
| Breast Cancer 1 | ROCAUC | 0.529 | 0.495 | 0.582 | 0.503 |
|  | PRAUC | 0.086 | 0.082 | 0.083 | 0.080 |
| Breast Cancer 2 | ROCAUC | 0.608 | 0.633 | 0.719 | 0.668 |
|  | PRAUC | 0.065 | 0.288 | 0.168 | 0.326 |
| Breast Cancer 3 | ROCAUC | 0.537 | 0.552 | 0.719 | 0.556 |
|  | PRAUC | 0.348 | 0.338 | 0.357 | 0.337 |
| Colorectal Cancer | ROCAUC | 0.558 | 0.567 | 0.636 | 0.557 |
|  | PRAUC | 0.170 | 0.191 | 0.175 | 0.152 |
| Lung Cancer 1 | ROCAUC | 0.426 | 0.384 | 0.684 | 0.355 |
|  | PRAUC | 0.925 | 0.913 | 0.919 | 0.915 |
| Lung Cancer 2 | ROCAUC | 0.499 | 0.534 | 0.660 | 0.496 |
|  | PRAUC | 0.540 | 0.583 | 0.597 | 0.545 |
| Lung Cancer 3 | ROCAUC | 0.571 | 0.536 | 0.857 | 0.357 |
|  | PRAUC | 0.741 | 0.735 | 0.909 | 0.699 |

Table 12: Ablation of model output logit Standard Deviations over all datasets. "Logit Std (Per Patient) denotes the average standard deviation of logit outputs over all audited paraphrases of that patient. Logit Std (Overall) denotes the overall standard deviation of all model output logits. We see that in most cases, the Per Patient Std is an order of magnitude smaller than the overall Std. Lung Cancer 3 may be an exception, as it is the smallest dataset size by far.

| Trial Name | Logit Std (Per Patient) | Logit Std (Overall) |
|---|---|---|
| Breast Cancer 1 | 0.0086 | 0.029 |
| Breast Cancer 2 | 0.0036 | 0.012 |
| Breast Cancer 3 | 0.0330 | 0.164 |
| Colorectal Cancer | 0.0185 | 0.078 |
| Lung Cancer 1 | 0.0145 | 0.079 |
| Lung Cancer 2 | 0.0488 | 0.208 |
| Lung Cancer 3 | 0.1467 | 0.264 |

