# OpenReview forum: "MediTab: Scaling Medical Tabular Data Predictors via Data Consolidation, Enrichment, and Refinement"
_ICLR.cc/2024/Conference — Submitted to ICLR 2024_

### Official Review · Reviewer_bkYv · 2023-10-28

**Soundness:** 3 good
**Presentation:** 3 good
**Contribution:** 3 good
**Rating:** 6
**Confidence:** 3

**Summary:**

This work proposes an approach for developing learning prediction models using heterogenous tabular data sources and schemas and across tasks. The approach relies on large language models to represent structured data in natural language as a common representation across contexts, aggregate datasets from like and unlike tasks and populations, and do zero and few-shot prediction. The approach is applied to several medical data (primarily clinical trials with some retrospective observational data). The approach generally outperforms fully-supervised baselines and further performs well in zero and few-shot settings.

**Strengths:**

* The method appears to yield performant predictive models across datasets and tasks, and without requiring significant amounts of labeled target data.
* The approach to representing and harmonizing structured data in natural language using LLMs is general and plausibly would continue to work well outside of the context evaluated in this work.

**Weaknesses:**

* I have several concerns related to clarity and lack of detail given to some important aspects of the methodology and experiments. These are elaborated on in the Questions section below.
* The datasets chosen are relatively small and relatively low-dimensional (e.g. ~10s of fields at most). An area where this method might be useful is with tabular data of much greater dimensionality, as is typical in healthcare contexts.

**Questions:**

* How are the “External Patient Databases” used (MIMIC-IV and PMC-Patients)? Are they used as supplementary databases during the psuedolabeling step?
* How is the MIMIC-IV data processed? The information in Table 1 that shows that there are only 2 categorical, 1 binary, and 1 numerical feature in MIMIC-IV. As MIMIC-IV is much richer (potentially thousands of features) it is unclear which components of the database are actually used and no details are provided.
* In section 2.4, why is the initial model trained on all available training data from $T_1$ considered a multi-task model (designated by $f_{MTL}$)? If I understand correctly, this model is trained on one task, but several datasets.
* The description of the psuedolabeling step is not entirely clear to me. Is the idea to take the initial model for the target task, make predictions for the target task on data collected for other tasks, and then use those predictions as pseudo-labels for further training? This seems peculiar because it is not clear that this should fundamentally improve performance for the target task given that the pseudo-labels are essentially just predictions of the target label derived from information in the target task database(s).
* If available, it would be relevant to compare to baselines that pool over datasets with rule-based schema harmonization. For example, in the context of electronic health records and claims data, there are standards such as the OMOP Common Data Model that provide the means of mapping data from disparate sources to a shared schema.
* An ablation experiment that removes the auditing steps (both the LLM sanity check and the Data Shapley checks) and the pseudolabeling step would help gain insight into the marginal value that they provide, especially as they are positioned as the novel methodological contributions of this work relative to TabLLM (if I understand correctly).

---

> ### Author Response · Authors · 2023-11-19
> **Response to Reviewer bkYv (1/2)**
>
> Thank you for taking the time to review our paper and your valuable feedback! Here are our responses to each point in order:
>
> - How are the “External Patient Databases” used (MIMIC-IV and PMC-Patients)? Are they used as supplementary databases during the pseudo-labeling step?
>
> Exactly! In Step 2, MediTab retrieves samples from external databases and tries to annotate and audit the samples to build a clean, augmented dataset. That data will then be involved in enhancing the training of MediTab to obtain better performances.
>
> - How is the MIMIC-IV data processed?
>
> To simplify the setup, we use the simple baseline features of the patients, including their age and gender, and their clinical discharge notes as the external datasets to obtain shapley values and pseudolabels.
>
> - Is the idea to take the initial model for the target task, make predictions for the target task on data collected for other tasks, and then use those predictions as pseudo-labels for further training?
>
> Yes you are correct! The model is trained on the same outcome prediction (predicting binary patient survival), and we refer to this as different tasks as they come from very different clinical trial datasets. We have clarified this in the text in section 2.2 step 1. We primarily use the pseudolabels to facilitate training the zero-shot (no access to the original data) and are not meant to improve the performance of the original supervised model. Thank you for pointing out this point, and we have clarified this in section 2.2 step 3.

---

> ### Author Response · Authors · 2023-11-19
> **Response to Reviewer bkYv (2/2)**
>
> - An ablation experiment that removes the auditing steps and the pseudolabeling step would help gain insight
>
> This is a great point! We have added ablations regarding choice of downstream model (biobert vs clinicalbert vs bert), as well as further comparisons with TABLLM. We performed ablation on the percentile cutoff for the data-importance shapley score.
>
> Data importance ablation (Please see the Appendix G Table 11 on page 29! ): after each percentile was calculated, a single epoch of training was performed to obtain the ROC-AUC and PR-AUC. We see that empirically, using the 50 percentile cutoff performs the best. A small cutoff allows in too many irrelevant examples, and a high cutoff may remove too much diversity from the data.
>
> | Trial Name        | Metric | 10\%  | 25\%  | 50\%  | 90\%  |
> |-------------------|--------|-------|-------|-------|-------|
> | Breast Cancer 1   | ROCAUC | 0.529 | 0.495 | 0.582 | 0.503 |
> |                   | PRAUC  | 0.086 | 0.082 | 0.083 | 0.080 |
> | Breast Cancer 2   | ROCAUC | 0.608 | 0.633 | 0.719 | 0.668 |
> |                   | PRAUC  | 0.065 | 0.288 | 0.168 | 0.326 |
> | Breast Cancer 3   | ROCAUC | 0.537 | 0.552 | 0.719 | 0.556 |
> |                   | PRAUC  | 0.348 | 0.338 | 0.357 | 0.337 |
> | Colorectal Cancer | ROCAUC | 0.558 | 0.567 | 0.636 | 0.557 |
> |                   | PRAUC  | 0.170 | 0.191 | 0.175 | 0.152 |
> | Lung Cancer 1     | ROCAUC | 0.426 | 0.384 | 0.684 | 0.355 |
> |                   | PRAUC  | 0.925 | 0.913 | 0.919 | 0.915 |
> | Lung Cancer 2     | ROCAUC | 0.499 | 0.534 | 0.660 | 0.496 |
> |                   | PRAUC  | 0.540 | 0.583 | 0.597 | 0.545 |
> | Lung Cancer 3     | ROCAUC | 0.571 | 0.536 | 0.857 | 0.357 |
> |                   | PRAUC  | 0.741 | 0.735 | 0.909 | 0.699 |
>
> Different serialization strategies ablation (Appendix G Table 10, page 28)
>
> |     Trial Name    | Metric | Simple Text | Praphrase | Audited Paraphrase |
> |:-----------------:|:------:|:-----------:|:---------:|:------------------:|
> |  Breast Cancer 1  | ROCAUC |    0.607    |   0.620   |        0.617       |
> |                   |  PRAUC |    0.098    |   0.107   |        0.105       |
> |  Breast Cancer 2  | ROCAUC |    0.753    |   0.753   |        0.876       |
> |                   |  PRAUC |    0.083    |   0.083   |        0.135       |
> |  Breast Cancer 3  | ROCAUC |    0.760    |   0.758   |        0.764       |
> |                   |  PRAUC |    0.452    |   0.481   |        0.476       |
> | Colorectal Cancer | ROCAUC |    0.695    |   0.691   |        0.705       |
> |                   |  PRAUC |    0.259    |   0.264   |        0.256       |
> |   Lung Cancer 1   | ROCAUC |    0.699    |   0.737   |        0.717       |
> |                   |  PRAUC |    0.975    |   0.979   |        0.972       |
> |   Lung Cancer 2   | ROCAUC |    0.699    |   0.697   |        0.716       |
> |                   |  PRAUC |    0.679    |   0.680   |        0.715       |
> |   Lung Cancer 3   | ROCAUC |    0.607    |   0.893   |        0.929       |
> |                   |  PRAUC |    0.697    |   0.957   |        0.968       |
>
>
> Our results indicated that the basic approach of using "column name: column value," similar to what's demonstrated in TabLLM, was effective. However, we also observed that enhancing this format with paraphrased examples led to better performance. Furthermore, we find that audited examples improve performance the most. We believe that this performance benefit is useful and serves to justify our usage of more advanced paraphrasing and auditing techniques to address our points of addressing model hallucinations and data augmentation.
>
> We hope that these revisions help address some of your concerns regarding the paper, and look forward to further discussion!

---

> > ### Comment · Reviewer_bkYv · 2023-11-20
> >
> > Thank you for the clarification and running those additional ablation studies.
> >
> > I am planning to keep my overall score of 6 (weak accept) the same and increase my presentation sub-score to 3.
> >
> > I generally agree with reviewer R9Te that the ability for MediTab to leverage information from multiple training datasets and tasks (i.e. multiple "domains") to a plausible explanation for its performance improvement over baselines. To be clear, I don't consider this to be a weakness. However, if this hypothesis were to be tested, it's important to evaluate against other approaches to leveraging data from multiple disparate datasets.
> >
> > A couple of follow-up questions:
> >   * How should we interpret the low performance across board for the data importance ablation? Is this just because only one epoch of training was performed? Given the large difference in performance between these results and those reported elsewhere in the paper, do you expect those results to be informative in terms of selecting the best threshold?
> >   * Could you please add the description of the variables selected for each dataset to the paper, including MIMIC-IV, so that the procedure could be reproduced.

---

> > > ### Author Response · Authors · 2023-11-22
> > > **Response to Reviewer bkYv**
> > >
> > > We initially thought that one epoch (due to time constraints) would be enough to see the performance difference between the Shapley scores, as it is a fair comparison with the only difference being the cutoff. However, we agree with the reviewer that fully training the models for the full number of epochs in the real experiments is the best way to go.
> > >
> > > The following is a fully retrained version of the model trained only on the pseudo-labeled data, with varying levels of shapley score cutoff! We see that the trend of 50% cutoff still being the best persists, consistent with the single epoch approach. We have updated Table 11 accordingly.
> > >
> > > | Trial Name        | Metric | 10\%  | 25\%  | 50\%  | 90\%  |
> > > |-------------------|--------|-------|-------|-------|-------|
> > > | Breast Cancer 1   | ROCAUC | 0.529 | 0.495 | 0.582 | 0.503 |
> > > |                   | PRAUC  | 0.086 | 0.082 | 0.083 | 0.080 |
> > > | Breast Cancer 2   | ROCAUC | 0.608 | 0.633 | 0.719 | 0.668 |
> > > |                   | PRAUC  | 0.065 | 0.288 | 0.168 | 0.326 |
> > > | Breast Cancer 3   | ROCAUC | 0.537 | 0.552 | 0.719 | 0.556 |
> > > |                   | PRAUC  | 0.348 | 0.338 | 0.357 | 0.337 |
> > > | Colorectal Cancer | ROCAUC | 0.558 | 0.567 | 0.636 | 0.557 |
> > > |                   | PRAUC  | 0.170 | 0.191 | 0.175 | 0.152 |
> > > | Lung Cancer 1     | ROCAUC | 0.426 | 0.384 | 0.684 | 0.355 |
> > > |                   | PRAUC  | 0.925 | 0.913 | 0.919 | 0.915 |
> > > | Lung Cancer 2     | ROCAUC | 0.499 | 0.534 | 0.660 | 0.496 |
> > > |                   | PRAUC  | 0.540 | 0.583 | 0.597 | 0.545 |
> > > | Lung Cancer 3     | ROCAUC | 0.571 | 0.536 | 0.857 | 0.357 |
> > > |                   | PRAUC  | 0.741 | 0.735 | 0.909 | 0.699 |
> > >
> > > We have also added the additional info regarding the mimic dataset. Please refer to Appendix D Page 24 in the updated manuscript.

---

### Official Review · Reviewer_R9Te · 2023-11-01

**Soundness:** 3 good
**Presentation:** 3 good
**Contribution:** 1 poor
**Rating:** 3
**Confidence:** 5

**Summary:**

The manuscript proposes a framework utilizing LLM to perform alignment between different datasets for the same and different tasks. The framework prompts ChatGPT to summarize each row in a table into text and utilizes BioBERT as a classification model that takes text as input. Moreover, it trains an init model to annotate data from datasets of other tasks and clean such data with Shapley scores into supplementary data samples.

**Strengths:**

- The paper is well-written and easy to follow.

**Weaknesses:**

- The framework is quite straightforward, and there is not much technical contribution. It is mostly a combination of multiple existing models. And the idea of transferring tabular data into text is not novel at all. There are a bunch of existing works [1][2][3], including one of their baselines TabLLM[4]. The further incorporation of text information from samples from other datasets is just one trivial step forward. Furthermore, [4] actually proved that a template for transferring the tabular data works better than an LLM. Yet, in this paper, there is no comparison for such serialization methods.

- The author didn’t specify what exact features are included in these experimental datasets. Also, it is unclear how many columns are overlapped between different datasets. Yet, if there is a large portion of feature overlapping, maybe simple concatenation and removing or recoding of the missing columns will work just as well. There is no discussion regarding this whatsoever.

- The step 2 in section 2.2 is confusing:
    - The authors claimed that they used active learning in step 2. Is the “active learning pipeline” method the same as traditional active learning that select informative samples to label? If not, the description can mislead the readers.
    - The authors claimed that they cleaned supplementary dataset T_{1, sup} with a data audit module based on data Shapley scores. More experiments are expected to demonstrate the effectiveness of the audit module. Moreover, it would be better if the authors conducted more ablation studies to show whether the supplementary dataset improve the prediction performance.

- The datasets in Table 1 contain less than 3000 patients. It is very easy for the LLMs (e.g., BioBERT) to overfit the training set. It is unclear how the authors prevent overfitting during the fine-tuning phase.

- In Table 3, the proposed MediTab exhibits the capability to access multiple datasets during its training, in contrast to the other baseline models, which are constrained to employing a single dataset. This discrepancy in data utilization introduces an element of unfairness in the comparison. It would be more appropriate to conduct a comparison against models that have undergone training on multiple datasets. For instance, TabLLM, being a large language model, can readily undertake multi-dataset training with minor adjustments to its data preprocessing procedures. Therefore, a more equitable comparison would involve evaluating MediTab and TabLLM under identical conditions, both in the context of training on a single dataset and across multiple datasets.

- Most medical data, like MIMIC-IV, includes timestamp information of the patients’ multiple visits or collections. This framework completely ignores this part of the medical data, which limits their application to real-world clinical environments.

Reference:
1. Bertsimas, Dimitris & Carballo, Kimberly & Ma, Yu & Na, Liangyuan & Boussioux, Léonard & Zeng, Cynthia & Soenksen, Luis & Fuentes, Ignacio. (2022). TabText: a Systematic Approach to Aggregate Knowledge Across Tabular Data Structures. 10.48550/arXiv.2206.10381.
2. Yin, Pengcheng & Neubig, Graham & Yih, Wen-tau & Riedel, Sebastian. TaBERT: Pretraining for Joint Understanding of Textual and Tabular Data. ACL 2020.
3. Li, Y., Li, J., Suhara, Y., Doan, A., and Tan, W.-C. (2020). Deep entity matching with pre-trained language models. Proc. VLDB Endow., 14(1):50–60.
4. Stefan Hegselmann, Alejandro Buendia, Hunter Lang, Monica Agrawal, Xiaoyi Jiang, and David Sontag. Tabllm: Few-shot classification of tabular data with large language models. arXiv preprint arXiv:2210.10723, 2022.

**Questions:**

1. All questions in the above section.

2. Are there any overlaps of columns between the tabular data for the same tasks? Is it hard to do a simple concatenation? What’s the traditional method for dealing with the missing columns? Are they applicable to this situation?

3. For the choice of BioBERT and the QA model for salinity check, the author did not provide a reason for choosing these models.

---

> ### Author Response · Authors · 2023-11-19
> **Response to Reviewer R9Te (1/3)**
>
> Thank you for taking the time to review our paper and your valuable feedback! Here are our responses to each point in order:
>
> - The framework is quite straightforward..
>
> Thanks for pointing out the potential confusion about the difference between our method and previous papers. In TabLLM [1], the authors explore a suite of serialization strategies on nine public tabular prediction datasets and find the text template method works the best, better than the table-to-text variant. However, we would like to highlight that we propose two important aspects for the serialization:
>
> We account for the potential *errors* (or *hallucinations*) that may occur during the table-to-text translation process (called *paraphrasing* in this paper), which we found to undermine the performance of the models because wrong information or noises are added to the translated data. We propose a new *auditing* process to control this type of error.
> We also propose to distill the knowledge from large language models by augmenting the tabular data in this serialization process via multiple paraphrasing.
>
> Empirically, we found that both the above two aspects are crucial to the superior performance of MediTab, as shown in the table added to Appendix G Table 10, page 27:
>
> |     Trial Name    | Metric | Simple Text | Praphrase | Audited Paraphrase |
> |:-----------------:|:------:|:-----------:|:---------:|:------------------:|
> |  Breast Cancer 1  | ROCAUC |    0.607    |   0.620   |        0.617       |
> |                   |  PRAUC |    0.098    |   0.107   |        0.105       |
> |  Breast Cancer 2  | ROCAUC |    0.753    |   0.753   |        0.876       |
> |                   |  PRAUC |    0.083    |   0.083   |        0.135       |
> |  Breast Cancer 3  | ROCAUC |    0.760    |   0.758   |        0.764       |
> |                   |  PRAUC |    0.452    |   0.481   |        0.476       |
> | Colorectal Cancer | ROCAUC |    0.695    |   0.691   |        0.705       |
> |                   |  PRAUC |    0.259    |   0.264   |        0.256       |
> |   Lung Cancer 1   | ROCAUC |    0.699    |   0.737   |        0.717       |
> |                   |  PRAUC |    0.975    |   0.979   |        0.972       |
> |   Lung Cancer 2   | ROCAUC |    0.699    |   0.697   |        0.716       |
> |                   |  PRAUC |    0.679    |   0.680   |        0.715       |
> |   Lung Cancer 3   | ROCAUC |    0.607    |   0.893   |        0.929       |
> |                   |  PRAUC |    0.697    |   0.957   |        0.968       |
>
>
> This work builds upon TabLLM by including data auditing of the paraphrased results as well as using external datasets for medical tabular prediction.
>
> Our results indicated that the basic approach of using "column name: column value," similar to what's demonstrated in TabLLM, was effective. However, we also observed that enhancing this format with paraphrased examples led to better performance. Furthermore, we find that audited examples improve performance the most. We believe that this performance benefit is useful and serves to justify our usage of more advanced paraphrasing and auditing techniques to address our first 2 points of addressing model hallucinations and data augmentation.
>
> [1] Hegselmann S, Buendia A, Lang H, et al. Tabllm: Few-shot classification of tabular data with large language models[C]//International Conference on Artificial Intelligence and Statistics. PMLR, 2023: 5549-5581.
>
>
> - The author didn’t specify what exact features are included in these experimental datasets…
>
> We have added more information regarding the dataset columns in Table 5, Appendix C.2, Page 17 and a discussion.
>
> |     Trial Name    | # Vars Shared | # Vars Total |
> |:-----------------:|:-------------:|:------------:|
> |  Breast Cancer 1  |       4       |      15      |
> |  Breast Cancer 2  |       15      |      48      |
> |  Breast Cancer 3  |       7       |      18      |
> | Colorectal Cancer |       5       |      16      |
> |   Lung Cancer 1   |       1       |      17      |
> |   Lung Cancer 2   |       6       |      41      |
> |   Lung Cancer 3   |       7       |      26      |
>
> The columns are all quite diverse in terms of semantic meaning, and we believe that traditional methods like data imputation or renaming/removing would not apply here, as we have fundamentally different features.
>
> - Is the “active learning pipeline” method the same as traditional active learning that select informative samples to label?
>
> Thank you for pointing this out! Step 2 (Page 3) in the pipeline is conceptually similar to the active learning process. First, the model retrieves samples from other datasets that have different target variables to predict. Second, the model annotates the retrieved samples to build a noisy-label dataset. Third, the model runs the data audit process to select the samples that are most likely to be annotated correctly. Nonetheless, we agree that this was not the best choice of wording and have rephrased the sentence.

---

> > ### Author Response · Authors · 2023-11-19
> > **Response to Reviewer R9Te (2/3)**
> >
> > - More experiments are expected to demonstrate the effectiveness of the audit module.
> >
> > The external datasets were annotated with the pre-trained model and audits with our pipeline. Then, these clean pseudo-label data are used to fine-tune a model for the zero-shot and few-shot settings.
> >
> > Data importance ablation (Please see the Appendix G Table 11 on page 29! ): after each percentile was calculated, a single epoch of training was performed to obtain the ROC-AUC and PR-AUC. We see that empirically, using the 50 percentile cutoff performs the best. A small cutoff allows in too many irrelevant examples, and a high cutoff may remove too much diversity from the data.
> >
> > | Trial Name        | Metric | 10\%  | 25\%  | 50\%  | 90\%  |
> > |-------------------|--------|-------|-------|-------|-------|
> > | Breast Cancer 1   | ROCAUC | 0.485 | 0.505 | 0.500 | 0.445 |
> > |                   | PRAUC  | 0.072 | 0.082 | 0.092 | 0.061 |
> > | Breast Cancer 2   | ROCAUC | 0.591 | 0.426 | 0.728 | 0.656 |
> > |                   | PRAUC  | 0.275 | 0.025 | 0.082 | 0.168 |
> > | Breast Cancer 3   | ROCAUC | 0.476 | 0.518 | 0.525 | 0.517 |
> > |                   | PRAUC  | 0.186 | 0.207 | 0.234 | 0.210 |
> > | Colorectal Cancer | ROCAUC | 0.517 | 0.497 | 0.489 | 0.466 |
> > |                   | PRAUC  | 0.139 | 0.132 | 0.135 | 0.118 |
> > | Lung Cancer 1     | ROCAUC | 0.429 | 0.458 | 0.535 | 0.391 |
> > |                   | PRAUC  | 0.931 | 0.929 | 0.898 | 0.917 |
> > | Lung Cancer 2     | ROCAUC | 0.514 | 0.536 | 0.573 | 0.512 |
> > |                   | PRAUC  | 0.472 | 0.452 | 0.439 | 0.438 |
> > | Lung Cancer 3     | ROCAUC | 0.464 | 0.500 | 0.671 | 0.607 |
> > |                   | PRAUC  | 0.766 | 0.745 | 0.812 | 0.765 |
> >
> >
> > - The datasets in Table 1 contain less than 3000 patients. It is very easy for the LLMs (e.g., BioBERT) to overfit the training set. It is unclear how the authors prevent overfitting during the fine-tuning phase.
> >
> > The sample size is small due to the nature of clinical trials, where even a phase III trial may have at most 1K patients, and phase I and II trials usually have less than 100. That is the reason why we propose MediTab to augment the model with external datasets, e.g., from EHRs. Technically, we made train/val/test splits to make sure there was no data leakage. In experiments, we choose the model with the best performance on the validation set. The performance on the test set hence shows the generalization capability. Since the datasets are not large, we decided to only train the model for a small number of epochs to not overfit: with 3 epochs for pre-training and 1 epoch for fine-tuning. We have added the discussion to section 2.5 page 5.
> >
> > - a more equitable comparison would involve evaluating MediTab and TabLLM under identical conditions, both in the context of training on a single dataset and across multiple datasets.
> >
> > We agree that TabLLM is capable of pretraining across multiple datasets like our method, but we also believe in preserving the integrity of its original design, as it was not created for the purpose of training across multiple datasets. However, we have performed an ablation regarding single dataset training, and we see that TabLLM struggles to perform on our domain-specific clinical datasets.
> >
> > - Most medical data, like MIMIC-IV, includes timestamp information of the patients’ multiple visits or collections. This framework completely ignores this part of the medical data, which limits their application to real-world clinical environments.
> >
> > We agree that extending to longitudinal data is crucial. Since the focus of this work is tabular data prediction, we will explore it in future work. However, while we do not explicitly consider sequential patient data in this work, only traditional tabular data, we believe that this is a logical next step. In essence, we can structure sequential visits into longer textual descriptions, which can also be handled by BERT models. We believe that with appropriate adaptation, the ideas of paraphrasing, data auditing, and enrichment can also shine for sequential data with MediTab model.

---

> > > ### Author Response · Authors · 2023-11-19
> > > **Response to Reviewer R9Te (3/3)**
> > >
> > > - For the choice of BioBERT and the QA model for salinity check, the author did not provide a reason for choosing these models.
> > >
> > > We have added ablations regarding the choice of downstream model (BioBERT vs. ClinicalBERT vs. BERT), as well as further comparisons with TabLLM. Please see the Appendix G Table 9 on page 28!
> > >
> > > In detail, we compared NEW ablations of different base models (BERT, BioBERT, ClinicalBERT, and TabLLM) in terms of downstream performance, trained from scratch, respectively, as shown in the table below, about their average performance across seven datasets.
> > >
> > > | Trial Name        | Metric | BERT  | ClinicalBERT | BioBERT | TabLLM |
> > > |-------------------|--------|-------|--------------|---------|--------|
> > > | Breast Cancer 1   | ROCAUC | 0.588 | 0.581        | 0.591   | -      |
> > > |                   | PRAUC  | 0.097 | 0.082        | 0.094   | -      |
> > > | Breast Cancer 2   | ROCAUC | 0.485 | 0.724        | 0.803   | -      |
> > > |                   | PRAUC  | 0.023 | 0.026        | 0.060   | -      |
> > > | Breast Cancer 3   | ROCAUC | 0.696 | 0.734        | 0.721   | 0.616  |
> > > |                   | PRAUC  | 0.392 | 0.366        | 0.437   | 0.302  |
> > > | Colorectal Cancer | ROCAUC | 0.613 | 0.700        | 0.705   | -      |
> > > |                   | PRAUC  | 0.233 | 0.186        | 0.267   | -      |
> > > | Lung Cancer 1     | ROCAUC | 0.555 | 0.479        | 0.699   | -      |
> > > |                   | PRAUC  | 0.962 | 0.949        | 0.971   | -      |
> > > | Lung Cancer 2     | ROCAUC | 0.544 | 0.616        | 0.711   | 0.619  |
> > > |                   | PRAUC  | 0.483 | 0.616        | 0.691   | 0.562  |
> > > | Lung Cancer 3     | ROCAUC | 0.357 | 0.893        | 0.893   | 0.804  |
> > > |                   | PRAUC  | 0.695 | 0.957        | 0.957   | 0.826  |
> > >
> > >
> > > We observed that the model selection choice is similar for BioBERT and ClinicalBERT. However, TabLLM does not converge for 4 out of the 7 datasets. We believe this may be due to the small amount of training data, but further research should be done to investigate this behavior fully.
> > >
> > > In the second experiment, we performed a NEW ablation on the effect of different serialization strategies. We’ve added the results to Appendix G Table 10, page 27, in the new version.
> > >
> > > |     Trial Name    | Metric | Simple Text | Praphrase | Audited Paraphrase |
> > > |:-----------------:|:------:|:-----------:|:---------:|:------------------:|
> > > |  Breast Cancer 1  | ROCAUC |    0.607    |   0.620   |        0.617       |
> > > |                   |  PRAUC |    0.098    |   0.107   |        0.105       |
> > > |  Breast Cancer 2  | ROCAUC |    0.753    |   0.753   |        0.876       |
> > > |                   |  PRAUC |    0.083    |   0.083   |        0.135       |
> > > |  Breast Cancer 3  | ROCAUC |    0.760    |   0.758   |        0.764       |
> > > |                   |  PRAUC |    0.452    |   0.481   |        0.476       |
> > > | Colorectal Cancer | ROCAUC |    0.695    |   0.691   |        0.705       |
> > > |                   |  PRAUC |    0.259    |   0.264   |        0.256       |
> > > |   Lung Cancer 1   | ROCAUC |    0.699    |   0.737   |        0.717       |
> > > |                   |  PRAUC |    0.975    |   0.979   |        0.972       |
> > > |   Lung Cancer 2   | ROCAUC |    0.699    |   0.697   |        0.716       |
> > > |                   |  PRAUC |    0.679    |   0.680   |        0.715       |
> > > |   Lung Cancer 3   | ROCAUC |    0.607    |   0.893   |        0.929       |
> > > |                   |  PRAUC |    0.697    |   0.957   |        0.968       |
> > >
> > >
> > > Our results indicated that the basic approach of using "column name: column value," similar to what's demonstrated in TabLLM, was effective. However, we also observed that enhancing this format with paraphrased examples led to better performance. Furthermore, we find that audited examples improve performance the most. We believe that this performance benefit is useful and serves to justify our usage of more advanced paraphrasing and auditing techniques to address our points of addressing model hallucinations and data augmentation. For the QA model, we chose to use the most powerful version of UnifiedQA, a popular and effective model for general QA tasks, that we could reasonably run on all the paraphrased datasets. Although we explored using larger models such as LlaMA2, we found that running these models would take too much time for it to be practical.
> > >
> > > We hope that these revisions help address some of your concerns regarding the paper, and look forward to further discussion!

---

> ### Author Response · Authors · 2023-11-22
> **Kindly reminder of the end of the rebuttal period**
>
> Dear Reviewer R9Te,
>
> As the discussion period is coming to an end tomorrow, we kindly ask you to review our response to your comments and let us know if you have any further queries. Alternatively, if you could raise the score of the paper, we would be extremely grateful. We eagerly anticipate your response and are committed to addressing any remaining concerns before the discussion period concludes.
>
> Best regards,
>
> Authors

---

> > ### Comment · Reviewer_R9Te · 2023-11-22
> >
> > Thank you for your response and the appended experiments.
> > - The main contribution of the paper is the proposed audit paraphrase. The authors sent patients’ data to LLM (OpenAI’s API) to generate text, which raises the increased risk of patient data leakage. Is this allowed by the Cancer Research Platform DUA? (Physionet prohibits sharing access to the data with third parties, including sending it through APIs provided by companies like OpenAI - https://physionet.org/news/post/415) Moreover, the model cannot handle the temporal nature of clinical tabular data, which significantly limits its applications to real clinical environments.
> > - It is unclear whether the authors use the validation dataset (e.g., in Table 1, 2). What does it mean “training and validating the best model on training data” in Section 2.5? How did the authors determine that 3 epochs for pretraining and 1 epoch for finetuning were optimal without validation sets? Are the optimal number of epochs the same in various tasks? How to use the model (e.g., to find the best number of epochs for pretraining and finetuning) for new datasets and tasks?
> > - The inconsistency in generated texts from multiple runs with LLM raises concerns about its impact on model output. Experimentation results (e.g., in Table 3, 4) are expected to exhibit standard deviations. Additionally, it would be beneficial for the authors to verify whether the model predicts consistent risks for different texts generated from the same patients.
> > - Besides, it is very straightforward to apply TabLLM to multiple dataset settings. For instance, representing each patient's data as text and utilizing TabLLM for training and testing is a straightforward process, independent of the dataset source. It would be better if the authors compared MediTab with TabLLM in the same settings (i.e., with multiple datasets during training).
> > - “Active learning” still appears in Figure 1 of the updated version.
> >
> > Overall, the paper can still be further improved and I will keep my score.

---

> > > ### Author Response · Authors · 2023-11-22
> > > **Response to Reviewer R9Te**
> > >
> > > - Patient data leakage
> > >
> > > Thanks for pointing out this crucial concern. We clarify that we did not release any physionet data to OpenAI’s API because we do not augment these samples from the MIMIC-IV database, similarly for PMC-patients. Instead, we only augment the synthetic patient trial samples that are built based on the raw patient records encoded in clinical trial data models such as ADaM and SDTM. We expect the augmentation part can be replaced by local LLMs such as LLAMA2 after instruction tuning. We have clarified Appendix A Broader Impact (page 14) to reflect this.
> > >
> > > We agree that extending to longitudinal data is crucial. However, while we do not explicitly consider sequential patient data in this work, only traditional tabular data, we believe that this is a logical next step. In essence, we can structure sequential visits into longer textual descriptions, which can also be handled by BERT models. We believe that with appropriate adaptation, the ideas of paraphrasing, data auditing, and enrichment can also shine for sequential data with the MediTab model.
> > >
> > > - Optimal number of epochs
> > >
> > > Due to the small number of samples in some datasets, we thought it would be best to use all possible training samples in the learning phase without leaking information in the testing phase. As the label distribution is highly skewed (Table 1), it may also bias our model if validation samples were chosen randomly. In our specific circumstance, we chose to simply train the model for 3 epochs on all of the datasets and then perform a single pass of fine-tuning, without significant hyperparameter optimization, due to the small amount of data and the good performance that it gives. Further work will extend this method on larger datasets to fully investigate train/val/test splits. This clarification has been made in Section 2.5 (page 5).
> > >
> > > - Inconsistency in generated texts
> > >
> > > For the variant phrases of a single patient, the predictions made by the model will differ. However, we want  to clarify that (1) we augment every patient with different paraphrases, but the the model is trained to make consistent predictions for these variants; (2) in the testing phase, the augmentations also provide a way to inspect the uncertainty in the model's prediction and an ensemble prediction, which improves the robustness and reliability of the model.
> > >
> > > For standard deviations, we report randomized runs of MediTab in terms of ROC-AUC ranking on multiple runs in Figure 5.
> > >
> > > - Multi-dataset TabLLM
> > >
> > > We agree with this point, and have been running it in the background to compare TabLLM trained on the same train / test split as Meditab. We’ve added the results in the updated version.
> > >
> > > | Trial Name        | Metrics | XGBoost | MLP    | FT-Transformer | TransTab | TabLLM (Single) | TabLLM (Multisource) | MediTab |
> > > |-------------------|---------|---------|--------|----------------|----------|-----------------|----------------------|---------|
> > > | Breast Cancer 1   | AUROC   | 0.543   | 0.6091 | 0.5564         | 0.5409   | -               | -                    | 0.6182  |
> > > |                   | PRAUC   | 0.0796  | 0.0963 | 0.0803         | 0.0923   | -               | -                    | 0.1064  |
> > > | Breast Cancer 2   | AUROC   | 0.6827  | 0.6269 | 0.6231         | 0.6      | -               | -                    | 0.8397  |
> > > |                   | PRAUC   | 0.1559  | 0.1481 | 0.052          | 0.0365   | -               | -                    | 0.1849  |
> > > | Breast Cancer 3   | AUROC   | 0.6489  | 0.7065 | 0.6338         | 0.71     | 0.6163          | 0.6103               | 0.7529  |
> > > |                   | PRAUC   | 0.3787  | 0.4    | 0.3145         | 0.4133   | 0.3023          | 0.2977               | 0.4567  |
> > > | Colorectal Cancer | AUROC   | 0.6704  | 0.6337 | 0.5951         | 0.7096   | -               | -                    | 0.7107  |
> > > |                   | PRAUC   | 0.2261  | 0.1828 | 0.1541         | 0.2374   | -               | -                    | 0.2402  |
> > > | Lung Cancer 1     | AUROC   | -       | 0.6023 | -              | 0.6499   | -               | -                    | 0.7246  |
> > > |                   | PRAUC   | -       | 0.9555 | -              | 0.9672   | -               | -                    | 0.9707  |
> > > | Lung Cancer 2     | AUROC   | 0.6976  | 0.5933 | 0.6093         | 0.5685   | 0.6188          | 0.6279               | 0.6822  |
> > > |                   | PRAUC   | 0.6865  | 0.5662 | 0.5428         | 0.4922   | 0.5619          | 0.5772               | 0.671   |
> > > | Lung Cancer 3     | AUROC   | 0.6976  | 0.6429 | 0.5357         | 0.6786   | 0.8036          | 0.6786               | 0.8928  |
> > > |                   | PRAUC   | 0.7679  | 0.8501 | 0.725          | 0.7798   | 0.8256          | 0.7338               | 0.9478  |
> > >
> > > However, we see that without paraphrasing or auditing, TabLLM is unable to generalize across the highly varied domains, as they do not have many column name overlaps (See Table 5 on page 17).
> > >
> > > - Active learning in fig 1
> > >
> > > Fixed!

---

> > > > ### Comment · Reviewer_R9Te · 2023-11-22
> > > >
> > > > Thank you for your response.
> > > > - I am still not very clear about the implementation of this work. What do "synthetic patient trial samples" refer to? Is this work conducted on real patient data or synthetic patient data? How to generate text based on the patients' data if the raw data are not sent to OpenAI’s API?
> > > > - About the main contribution - What is the main difference between TabLLM (Multisource) and MediTab (Simple Text)? When removing paraphrasing and audit paraphrasing in MediTab, is the model similar to TabLLM (Multisource)? If yes, why does the model performance vary a lot (for example, TabLLM (Multisource): 0.610 in Table 3 and MediTab (Simple Text): 0.760 in Table 10 on AUROC for Breast Cancer 3 dataset)?
> > > > - For the learning part - directing using 3 epochs for training and 1 epoch for finetuning is not rigorous, especially for those baselines. How did the author select hyperparameters for the baselines? If they are selected in the same way, the experiment results are not convincing. The training of baselines might be suboptimal without parameter optimization.
> > > > - Inconsistency in generated texts: More experiments are expected to demonstrate the reliability of the model (for example, whether the model will generate the same risk with various texts from the same patients).
> > > > - In Table 3, why TabLLM(Multisource) doesn't converge, but MediTab does. Are they feeding with the same amount of training samples in the comparisons?

---

> > > > > ### Author Response · Authors · 2023-11-22
> > > > > **Response to Reviewer R9Te**
> > > > >
> > > > > We appreciate your prompt reply. We want to further clarify more to resolve your concerns.
> > > > >
> > > > > - "synthetic patient trial samples", real patient data or synthetic patient data, how to generate text?
> > > > >
> > > > > Sorry for any confusion in the previous response. As shown in Table 1 (Page 6), there are two types of patient datasets used in the paper: (1) dataset#1: clinical trial patient data and $T$ (2) dataset#2: the external patient database. Refer to Figure 2 on page 3, the dataset#2 is used by MediTab in Step 2: Learn, Annotate, and Audit, to get the cleaned data $T_{sup}$ as the enrichment for the target task dataset#1, where no sample was sent to LLM for augmentation. Only samples in dataset #1 are consolidated and augmented by LLM. The model was then pre-trained on $T_{sup}$ and then fine-tuned on $T$.
> > > > >
> > > > > To avoid leaking individual patient records from dataset #1, we did not send the raw clinical trial patient records to OpenAI but made a synthetic version of them. Technically, we used an KNN-based algorithm to generate tabular patient samples grounded on the real patient data, referring to [1]. In the future, we will develop an in-house LLaMA2 paraphraser to avoid the privacy issue. We have added this description to the updated version, in Appendix A on page 14.
> > > > >
> > > > > [1] Beigi M, Shafquat A, Mezey J, et al. Synthetic Clinical Trial Data while Preserving Subject-Level Privacy[C]//NeurIPS 2022 Workshop on Synthetic Data for Empowering ML Research. 2022.
> > > > >
> > > > > - Main difference between TabLLM (Multisource) and MediTab (Simple Text)
> > > > >
> > > > > The main differences of the two models are:
> > > > > TabLLM (Multisource) made a generative training on the linear serialized tabular samples aggregated from multiple datasets, based on bigscience/T0 model. It can also be developed with an GPT model, such as given tabular text descriptions, generating the labels as the prediction.
> > > > > MediTab (Simple Text) used discriminative training based on bidirectional transformers such as BERT to make classification by attaching the text encoder with a classifier.
> > > > > Such that, in our experiments, we observed generative training for tabular prediction is inferior to discriminative training in the target use cases.
> > > > >
> > > > > - How did the author select hyperparameters for the baselines
> > > > >
> > > > > While we did not make heavy hyperparameter tuning for MediTab, we did hyperparameter selection for the other baselines. Appendix E shows the baseline hyperparameter tuning information. Thank you for pointing this out, and we have added explicit reference to the hyperparameter tuning in the main text.
> > > > >
> > > > > - Inconsistency in generated texts
> > > > >
> > > > > Thank you for pointing this out, we were able to run another ablation regarding standard deviations of model output logits per patient vs overall output logit standard deviations, over all augmented texts. This is reflected in a new Table added to Appendix H Ablations, Page 29.
> > > > >
> > > > > |     Trial Name    | Logit Std (Per Patient) | Logit Std (Overall) |
> > > > > |:-----------------:|:-----------------------:|:-------------------:|
> > > > > |  Breast Cancer 1  |          0.0086         |        0.029        |
> > > > > |  Breast Cancer 2  |          0.0036         |        0.012        |
> > > > > |  Breast Cancer 3  |          0.0330         |        0.164        |
> > > > > | Colorectal Cancer |          0.0185         |        0.078        |
> > > > > |   Lung Cancer 1   |          0.0145         |        0.079        |
> > > > > |   Lung Cancer 2   |          0.0488         |        0.208        |
> > > > > |   Lung Cancer 3   |          0.1467         |        0.264        |
> > > > >
> > > > > Ablation of model output logit Standard Deviations over all datasets. "Logit Std (Per Patient) denotes the average standard deviation of logit outputs over all audited paraphrases of that patient. Logit Std (Overall) denotes the overall standard deviation of all model output logits. We see that in most cases, the Per Patient Std is an order of magnitude smaller than the overall Std. Lung Cancer 3 may be an exception, as it is the smallest dataset size by far.
> > > > >
> > > > > - In Table 3, why TabLLM (Multisource) doesn't converge, but MediTab does. Are they feeding with the same amount of training samples in the comparisons?
> > > > >
> > > > > Yes, they are fed with the same number of patients. However, as we discussed above, two methods are based on different training strategies. In our experiments, we find generative training for tabular prediction is generally less robust and may require larger and homogeneous samples to work well, as observed that TabLLM fails to converge on many datasets.

---

### Official Review · Reviewer_zDeD · 2023-11-02

**Soundness:** 3 good
**Presentation:** 3 good
**Contribution:** 3 good
**Rating:** 6
**Confidence:** 3

**Summary:**

This paper proposes to scale medical tabular data predictors (MediTab) to handle diverse tabular inputs with varying features. The approach involves utilizing a large language models (LLMs) to merge tabular datasets, addressing the challenges presented by tables with different structures.  Additionally, it establishes a process for aligning out-of-domain data with the specific target task through a "learn, annotate, and refinement" pipeline.

**Strengths:**

1. The core concept behind MediTab, involving the consolidation, enrichment, and refinement modules, is well-founded in its aim to improve the scalability of predictive models designed for medical tabular data.
2. Implementing a sanity check through Large Language Model (LLM) Reflection is particularly important in the medical domain.
3. The paper is clearly written; the quality is sound.
4. The empirical evaluation in the paper is strong; relevant baselines are considered.
5. The coverage of related work is extensive, with clear distinctions drawn from other studies.

**Weaknesses:**

1. It is not clear, how was splitting into train/val/test organised?
2. The statement “After the quality check step, we obtain the original task dataset $T$ and the supplementary dataset $T_{sup}$ and have two potential options for model training. The first is to combine both datasets for training, but we have found that this approach results in suboptimal performance.” is a bit unclear. Why it is suboptimal, could authors elaborate on this?
3. Are these datasets prone to missing values, which is a critical concern in the medical domain? If so, what would be the recommended strategy for handling these missing values?
4. Results on Ablation studies on Different Learning strategies are provided. Could authors provide  Ablation studies on the different model components?

**Questions:**

1. Could the authors elaborate more on the LLM sanity check, as well as the results and tests provided in Appendix C.4-C.5? It has been discussed that it is crucial to conduct thorough evaluations of LLMs in healthcare, with particular attention to aspects of safety, equity, and bias [a]. Could the authors provide their thoughts on why they believe their model satisfies these requirements?
2. Could authors provide more detailed explanation of empirical studies and address points in Weakness section?

a. Singhal et al., Large language models encode clinical knowledge. Nature 620, 172–180 (2023).

---

> ### Author Response · Authors · 2023-11-19
> **Response to Reviewer zDeD (1/2)**
>
> Thank you for taking the time to review our paper and your valuable feedback! Here are our responses to each point in order:
>
>  - It is not clear, how was splitting into train/val/test organised?
>
> Thank you for pointing this out, as this was accidentally left out during paper editing. We have added this info to the dataset description in Tables 1 and 2.
>
> - The statement “After the quality check step, we obtain the original task dataset and the supplementary dataset and have two potential options for model training. The first is to combine both datasets for training, but we have found that this approach results in suboptimal performance.” is a bit unclear. Why it is suboptimal, could authors elaborate on this?
>
> Of course, we found that it was suboptimal due to the empirical lower performance obtained via simple pretraining on both the supplemental and the original task dataset, as opposed to the approach we used in the paper, which is the pretrain on the supplemental + model’s noisy labels and then finetune on the original task dataset with the real labels. We hypothesize that this is due to the much larger amount of supplemental data vs the real data.
>
> - Are these datasets prone to missing values, which is a critical concern in the medical domain? If so, what would be the recommended strategy for handling these missing values?
>
> In tabular prediction, it is a common practice to deal with missing values in the inputs—the same is true in clinical patient outcome prediction. The proposed method can conveniently deal with missing values: it serializes a row in tabular data into sentences and models on the textual description of the sample.
>
> - Results on Ablation studies on Different Learning strategies are provided. Could authors provide Ablation studies on the different model components?
>
> This is a great point! We have added ablations regarding the choice of downstream model (BioBERT vs. ClinicalBERT vs. BERT), as well as further comparisons with TabLLM. Please see Appendix G Table 9 on page 27!
>
> In detail, we compared NEW ablations of different base models (BERT, BioBERT, ClinicalBERT, and TabLLM) in terms of downstream performance, trained from scratch, respectively, as shown in the table below, about their average performance across seven datasets.
>
> | Trial Name        | Metric | BERT  | ClinicalBERT | BioBERT | TabLLM |
> |-------------------|--------|-------|--------------|---------|--------|
> | Breast Cancer 1   | ROCAUC | 0.588 | 0.581        | 0.591   | -      |
> |                   | PRAUC  | 0.097 | 0.082        | 0.094   | -      |
> | Breast Cancer 2   | ROCAUC | 0.485 | 0.724        | 0.803   | -      |
> |                   | PRAUC  | 0.023 | 0.026        | 0.060   | -      |
> | Breast Cancer 3   | ROCAUC | 0.696 | 0.734        | 0.721   | 0.616  |
> |                   | PRAUC  | 0.392 | 0.366        | 0.437   | 0.302  |
> | Colorectal Cancer | ROCAUC | 0.613 | 0.700        | 0.705   | -      |
> |                   | PRAUC  | 0.233 | 0.186        | 0.267   | -      |
> | Lung Cancer 1     | ROCAUC | 0.555 | 0.479        | 0.699   | -      |
> |                   | PRAUC  | 0.962 | 0.949        | 0.971   | -      |
> | Lung Cancer 2     | ROCAUC | 0.544 | 0.616        | 0.711   | 0.619  |
> |                   | PRAUC  | 0.483 | 0.616        | 0.691   | 0.562  |
> | Lung Cancer 3     | ROCAUC | 0.357 | 0.893        | 0.893   | 0.804  |
> |                   | PRAUC  | 0.695 | 0.957        | 0.957   | 0.826  |
>
>
> We observed that the model selection choice is similar for BioBERT and ClinicalBERT. However, TabLLM does not converge for 4 out of the 7 datasets. We believe this may be due to the small amount of training data, but further research should be done to investigate this behavior fully.

---

> ### Author Response · Authors · 2023-11-19
> **Response to Reviewer zDeD (2/2)**
>
> In the second experiment, we performed a NEW ablation on the effect of different serialization strategies. We’ve added the results to Appendix G Table 10, page 28, in the new version.
>
> |     Trial Name    | Metric | Simple Text | Praphrase | Audited Paraphrase |
> |:-----------------:|:------:|:-----------:|:---------:|:------------------:|
> |  Breast Cancer 1  | ROCAUC |    0.607    |   0.620   |        0.617       |
> |                   |  PRAUC |    0.098    |   0.107   |        0.105       |
> |  Breast Cancer 2  | ROCAUC |    0.753    |   0.753   |        0.876       |
> |                   |  PRAUC |    0.083    |   0.083   |        0.135       |
> |  Breast Cancer 3  | ROCAUC |    0.760    |   0.758   |        0.764       |
> |                   |  PRAUC |    0.452    |   0.481   |        0.476       |
> | Colorectal Cancer | ROCAUC |    0.695    |   0.691   |        0.705       |
> |                   |  PRAUC |    0.259    |   0.264   |        0.256       |
> |   Lung Cancer 1   | ROCAUC |    0.699    |   0.737   |        0.717       |
> |                   |  PRAUC |    0.975    |   0.979   |        0.972       |
> |   Lung Cancer 2   | ROCAUC |    0.699    |   0.697   |        0.716       |
> |                   |  PRAUC |    0.679    |   0.680   |        0.715       |
> |   Lung Cancer 3   | ROCAUC |    0.607    |   0.893   |        0.929       |
> |                   |  PRAUC |    0.697    |   0.957   |        0.968       |
>
>
> Our results indicated that the basic approach of using "column name: column value," similar to what's demonstrated in TabLLM, was effective. However, we also observed that enhancing this format with paraphrased examples led to better performance. Furthermore, we find that audited examples improve performance the most. We believe that this performance benefit is useful and serves to justify our usage of more advanced paraphrasing and auditing techniques to address our points of addressing model hallucinations and data augmentation. For the QA model, we chose to use the most powerful version of UnifiedQA, a popular and effective model for general QA tasks, that we could reasonably run on all the paraphrased datasets. Although we explored using larger models such as LlaMA2, we found that running these models would take too much time for it to be practical.
>
> - Could the authors elaborate more on the LLM sanity check, as well as the results and tests provided in Appendix C.4-C.5? It has been discussed that it is crucial to conduct thorough evaluations of LLMs in healthcare, with particular attention to aspects of safety, equity, and bias. Could the authors provide their thoughts on why they believe their model satisfies these requirements?
>
> Absolutely! They check to ensure that ChatGPT’s paraphrasing is accurate, and we see that it sometimes is not, based on the results of our QA model. Despite this, we only use the publicly available, open-source, pre-trained models from Huggingface as our base models and baselines. This reduces the chance of data contamination and improves safety and equity in terms of access to our methods. We evaluate across diverse datasets with different groups of patients to ensure it doesn't perpetuate bias or disparities, and all datasets are obtained, with permission, from Project Data Sphere, Github, and clinicaltrials.gov (for trial outcome prediction), where anyone can obtain the data for research purposes. The only private model is ChatGPT, but that is a current research direction we are looking into to further increase transparency and reproducibility. We have also added this discussion in Appendix A.
>
> We hope that these revisions address some of your concerns regarding the paper, and look forward to further discussion!

---

> > ### Comment · Reviewer_zDeD · 2023-11-21
> >
> > Thank you for the clarification and for conducting those additional ablation studies. I appreciate your efforts and am planning to maintain my overall score as it stands.
> > I do have one follow-up question regarding the missing data. It would be interesting to understand how the performance of the approach is affected by different missing data ratios, such as 5%, 10%, or 20%, etc. While this is not a critical aspect for the current study, gaining insight into this could be valuable.

---

> > > ### Author Response · Authors · 2023-11-22
> > > **Response to Reviewer zDeD**
> > >
> > > We agree that this would be an interesting ablation, as MediTab is able to naturally handle missing columns due to the table-to-text paraphrasing (especially considering that paraphrasing is not hallucination free, as shown by our audit step). Furthermore, we see that in the zero-shot scenario, even in the case where the model does not see any original data, prediction performance is still quite good, demonstrating MediTab’s robustness. However, due to the time constraint, running full experiments on missing data is not possible, but we are taking steps to achieve this, and will add an ablation the next time we are able to edit the paper!

---

### Author Response · Authors · 2023-11-19
**General Response: New manuscript revision**

Thank you everyone to taking the time to review our paper!
We have made extensive changes to the paper in response to all reviewers (highlighted in red the new clarified section), adding new ablations and tables showing:
1. New experiments of baseline model choice (Appendix G Table 9, page 27)
2. Different serialization strategies (Appendix G Table 10, page 28)
3. Data importances evaluations (Appendix G Table 11, page 29)
4. Clarifications on the columns of all the datasets we used in Appendix C.2, starting on page 15.
5. (11/20) Discussion regarding the references proposed by AC (Introduction Page 1 and Related Work Page 9)

---

> ### Comment · Area_Chair_6nEu · 2023-11-20
> **additional references**
>
> Dear authors: Thank you for the responses to the reviewers. A small additional point: Please discuss the relevance of these two papers:
>
> GenHPF: General Healthcare Predictive Framework with Multi-task Multi-source Learning, https://arxiv.org/abs/2207.09858
>
> Unifying Heterogeneous Electronic Health Records Systems via Text-Based Code Embedding, https://proceedings.mlr.press/v174/hur22a.html

---

> > ### Author Response · Authors · 2023-11-20
> > **Response to AC**
> >
> > We appreciate your recommendation of these papers. Their relevance has been acknowledged in the introduction (page 1) and the related work (page 9) of the updated manuscript. Here, we make a comparative analysis detailing how Meditab aligns with and differs from the concepts presented in these works.
> >
> > 1. GenHPF: General Healthcare Predictive Framework with Multi-task Multi-source Learning
> >
> > *Similarities*
> >
> > - **Motivation**: Both works explore how to enhance medical prediction with multi-source datasets, as MediTab works on clinical trial patient outcome prediction and clinical trial outcome predictions, while GenHPF works on patient health risk predictions with EHRs;
> >
> > - **Text-based Encoding**: Both follow the practice of previous tabular learning papers such as TransTab [1] and TabLLM [2] that converts input data into textualized descriptions, to employ a text-based embedding method to encode diverse inputs varying in features
> >
> > *Differences*
> >
> > - **Goal**: Besides consolidating datasets with various schema from the same domain, MediTab actively aligns out-domain datasets with the target domain through the proposed data engine. For instance, GenHPF performs multi-task learning on multiple EHR datasets; Meditab, on the other hand, MediTab is not to build a multi-task model but to maximize the utility of the prediction model for the target domain. It sets clinical trial patient data as the target domain and then tries to align EHRs with the target domain through the proposed “learn-annotate-audit” pipeline.
> >
> > - **Method**: MediTab proposes to (1) distill the knowledge from generative large language models and control the hallucinations (2) distill the knowledge from out-of-domain datasets with data quality audit, to enhance data consolidation. As shown in Appendix G Table 10, page 28, the proposed augmentation-audit module contributes to improvements over the original text serialization.
> >
> >
> > 2. Unifying Heterogeneous Electronic Health Records Systems via Text-Based Code Embedding (DescEmb)
> >
> > *Similarities*
> >
> > - **Motivation**: Both papers enhance medical prediction with transfer learning across datasets.
> >
> > - **Text-based Encoding**: Both aim to align the semantics of samples from different sources based on a text-based encoding approach like TabLLM and TransTab.
> >
> > *Differences*
> >
> > - DescEmb aims to attach text descriptions to medical code to perform text-based encoding in order to allow EHR predictive modeling across datasets. By contrast, MediTab works on tabular data from clinical trials, which include more than medical codes. Also, similar to the comparison made above, the core contribution is that MediTab proposes knowledge distillation from LLM and from out-of-domain datasets, to enhance the prediction model for the target task.
> >
> >
> > [1] Wang Z, Sun J. Transtab: Learning transferable tabular transformers across tables[J]. Advances in Neural Information Processing Systems, 2022, 35: 2902-2915.
> >
> > [2] Hegselmann S, Buendia A, Lang H, et al. Tabllm: Few-shot classification of tabular data with large language models[C]//International Conference on Artificial Intelligence and Statistics. PMLR, 2023: 5549-5581.

---

> ### Comment · Area_Chair_6nEu · 2023-11-20
> **reviewers, please acknowledge the responses from the authors**
>
> Dear reviewers: Please read the replies from the authors carefully, and submit your reactions. Please be open-minded in deciding whether to change your scores for the submission, taking into account the explanations and additional results provided by the authors.
>
> Thank you!

---

### Meta-Review · Area_Chair_6nEu · 2023-12-15

**Metareview:**

Two reviewers are slightly positive (score 6) while one is definitely negative (score 3). As the area chair, my opinion is that the average of these scores is reasonable, and my personal evaluation tends negative. I agree with reviewer R9Te that the framework is straightforward, and the idea of transferring tabular data into text is not novel.

In "Response to Reviewer R9Te (1/3)" the authors have an accuracy table where their method is the last column, "Audited Paraphrase." This method is the most accurate for only 7 out of 14 rows. This is a very weak overall measure, but combined with the lack of novelty and the heuristic nature of the approach, the research contribution is small.

Small note: For the clinical trial datasets, the train/test split is chronological before and after 2015, which is good. But for the outcome datasets, the train/test split method is unspecified. If the split is based on random shuffling (not chronological and not by data source), then this will tend to exaggerate the accuracy of all methods.

**Justification For Why Not Higher Score:**

Not a major contribution.

**Justification For Why Not Lower Score:**

Reasonably thorough work.

---

### Decision · Program_Chairs · 2024-01-16

Reject